# To Trust Or Not To Trust Your Vision-Language Model's Prediction

## Abstract

Vision-Language Models (VLMs) have demonstrated strong capabilities in aligning visual and textual modalities, enabling a wide range of applications in multimodal understanding and generation. While they excel in zero-shot and transfer learning scenarios, VLMs remain susceptible to misclassification, often yielding confident yet incorrect predictions. This limitation poses a significant risk in safety-critical domains, where erroneous predictions can lead to severe consequences. In this work, we introduce **TrustVLM**, a training-free framework designed to address the critical challenge of estimating when VLM's predictions can be trusted. Motivated by the observed modality gap in VLMs and the insight that certain concepts are more distinctly represented in the image embedding space, we propose a novel confidence-scoring function that leverages this space to improve misclassification detection. We rigorously evaluate our approach across 17 diverse datasets, employing 4 architectures and 2 VLMs, and demonstrate state-of-the-art performance, with improvements of up to $51.87\%$ in AURC, $9.14\%$ in AUROC, and $32.42\%$ in FPR95 compared to existing baselines. By improving the reliability of the model without requiring retraining, TrustVLM paves the way for safer deployment of VLMs in real-world applications. The code is available in Supplementary Material.

## 1 Introduction

Recent advances in Vision-Language Models (VLMs) have substantially transformed the field of multimodal learning by integrating visual and textual information within a unified framework. Models such as CLIP (Radford et al., 2021) and SigLIP (Zhai et al., 2023) have been widely adopted for diverse tasks, including zero-shot classification (Zhou et al., 2022), cross-modal retrieval (Ma et al., 2022), and image captioning (Barraco et al., 2022). Trained on large-scale image-text datasets scraped from the web, these models learn rich and transferable representations. However, despite their substantial capabilities, VLMs often encounter critical limitations when applied in real-world settings. One pressing concern is misclassification, where the model produces a confident, yet incorrect, prediction that may appear both semantically plausible and visually aligned with the input. While much of the existing research has focused on improving the accuracy of VLMs outputs, the equally important issue of trustworthiness, that is, determining whether a prediction should be accepted or flagged for human review, remains largely underexplored. This challenge is particularly consequential in safety-critical domains (Sun et al., 2024; Dong et al., 2023) such as autonomous driving, medical diagnostics, and surveillance, where erroneous predictions can lead to severe outcomes.

The challenge of misclassification detection (MisD) has been widely studied in the context of unimodal vision models, with numerous approaches proposed, including confidence-based scoring (Hendrycks & Gimpel, 2017; Jiang et al., 2018), outlier exposure (Cheng et al., 2024; Zhu et al., 2023; Liu et al., 2025), and confidence learning (Corbière et al., 2019; Moon et al., 2020). However, these approaches often overlook the unique complexities of multimodal models, where the interaction between visual inputs and textual semantics introduces additional sources of uncertainty (Dong et al., 2025; 2024a). Recently, Nguyen et al. (Nguyen et al., 2025) proposed utilizing human-level concepts to detect misclassification of VLMs. However, their approach necessitates the construction of numerous concepts for each class through the use of large language models, which can be a demanding process. Although MisD and out-of-distribution (OOD) detection (Dong et al., 2024b; Li et al., 2024) share the similar goal of identifying problematic inputs for a trained model, they target fundamentally distinct challenges. MisD focuses primarily on identifying in-distribution samples that are incorrectly

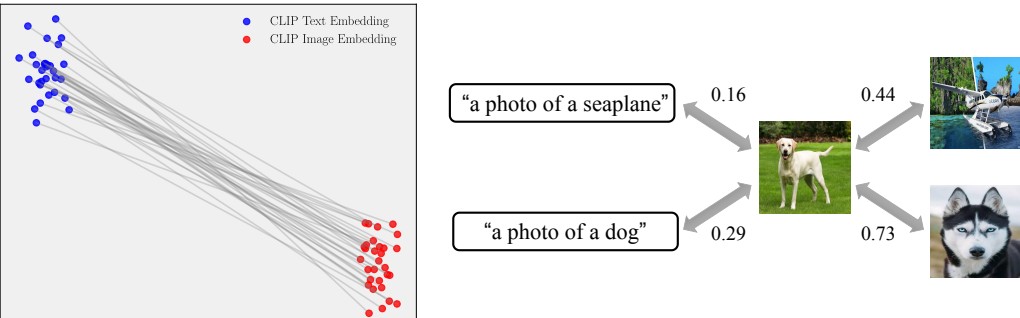

(a) CLIP embedding space    (b) Cosine similarity between image and text with CLIP embedding

Figure 1: (a) CLIP's image and text embeddings are located in two completely separate regions of the embedding space. (b) The concept of "dog" and "seaplane" is more distinguishable in the image embedding space than in the text embedding space. When using image-to-text similarity, the score difference between the concepts "dog" and "seaplane" is only **0.13** (0.29 − 0.16), making them less separable. In contrast, using image-to-image similarity yields a larger difference of **0.29** (0.73 − 0.44), indicating better separation between concepts within the image embedding space and potentially more reliable confidence estimation.

assigned to one of the known classes, often due to their proximity to decision boundaries or atypical feature representations within the learned data manifold. In contrast, OOD detection focuses on identifying inputs from entirely unseen distributions, representing novel or irrelevant stimuli rather than misclassifications within known classes. Consequently, methods tailored for one task often perform poorly on the other (Jaeger et al., 2022; Zhu et al., 2023).

To address the specific challenge of misclassification detection in VLMs, we propose **TrustVLM** – a training-free framework for evaluating the reliability of VLM predictions. Traditional zero-shot classification with VLMs relies primarily on the cosine similarity between text and image embeddings, often overlooking the structure and discriminative capacity of the image embedding space. This is a critical limitation, as previous work has shown a modality gap in VLMs like CLIP, where image and text embeddings reside in distinct regions of the shared representation space (Liang et al., 2022) (see Fig. 1). In particular, some concepts are more distinguishable in the image embedding space than in the text embedding space (Fig. 1 (b)). Building on this insight, TrustVLM leverages additional information from the image embedding space to design a novel confidence-scoring function for improved misclassification detection. Specifically, our framework employs an auxiliary vision encoder to store visual prototypes for each class and assess prediction reliability through image-to-image similarity with these prototypes. Beyond misclassification detection, these visual prototypes can also improve classification results on fine-grained datasets and be fine-tuned for improved downstream performance.

We conduct a rigorous evaluation of TrustVLM across 17 diverse datasets, 4 architectures, and 2 distinct VLMs. Our method achieves state-of-the-art performance in misclassification detection, with improvements of up to 51.87% in AURC, 9.14% in AUROC, and 32.42% in FPR95 over existing baselines. In addition, the use of visual prototypes improves the accuracy of fine-grained classification, giving an average improvement of 5.65%. The primary contributions of this work are as follows:

- We provide an empirical analysis of the limitations of existing MisD paradigms in VLMs, highlighting the value of leveraging information from the image embedding space.

- We propose TrustVLM, a training-free framework that combines image-to-text and image-to-image similarity to compute a robust confidence score for improved MisD.

- We show that visual prototypes not only support more reliable confidence estimation, but also improve fine-grained classification accuracy, and can optionally be fine-tuned for further gains.

- We extensively validate TrustVLM across datasets, model architectures, and VLMs, demonstrating its generality and effectiveness. Our source code will be made publicly available to support future research in MisD for VLMs.

## 2 PRELIMINARIES

**Vision-Language Models** typically comprise an image encoder that projects high-dimensional images into a low-dimensional embedding space and a text encoder that embeds natural language into a corresponding text embedding space. A prominent example is CLIP (Radford et al., 2021), trained on 400 million image-text pairs, which employs a contrastive loss to align image and text embeddings. Specifically, given a batch of image-text pairs, CLIP maximizes the cosine similarity for the matched pairs while minimizing it for unmatched ones. During inference, the class names of a target dataset are embedded using the text encoder with a prompt of the form "a photo of a [CLASS]", where [CLASS] is replaced with specific class names. The text encoder then generates text embeddings $\mathbf{t}_c$ for each class $c \in \mathcal{Y} = \{1, 2, \dots, C\}$, and the prediction probability for an input image $\boldsymbol{x}$ with embedding $\mathbf{f}_x$ is computed as:

$$p(y = \hat{y}|\boldsymbol{x}) = \frac{\exp\left(\cos\left(\mathbf{f}_x, \mathbf{t}_{\hat{y}}\right)/\tau\right)}{\sum_{c=1}^{C} \exp\left(\cos\left(\mathbf{f}_x, \mathbf{t}_c\right)/\tau\right)}, \tag{1}$$

where $\cos(\cdot, \cdot)$ denotes cosine similarity and $\tau$ is a temperature parameter. The final prediction for $\boldsymbol{x}$ is $\hat{y} = \arg\max_{y \in \mathcal{Y}} p(y|\boldsymbol{x})$, where $\hat{y}$ can be either correctly classified or misclassified.

**Misclassification Detection**, also known as failure detection (Corbière et al., 2019), serves as a critical safeguard for the reliable deployment of machine learning models in real-world applications. Its primary objective is to distinguish between correctly and incorrectly classified predictions, typically by leveraging confidence scores. Formally, let $\kappa$ denote a confidence-scoring function that quantifies the confidence of the model in its prediction. Given a threshold $\delta \in \mathbb{R}^+$, a decision function $g$ can be defined to detect misclassifications based on whether the confidence score exceeds this threshold. For a given input $\boldsymbol{x}$:

$$g(\boldsymbol{x}) = \begin{cases} \text{correct} & \text{if } \kappa(\boldsymbol{x}) \geq \delta, \\ \text{misclassified} & \text{otherwise.} \end{cases} \tag{2}$$

**Baselines for MisD of VLMs.** Given the prediction from Eq. (1), Maximum Softmax Probability (Hendrycks & Gimpel, 2017) can be readily computed as a confidence-scoring function. For a given input $\boldsymbol{x}$, MSP is defined as $\kappa(\boldsymbol{x}) = \max_{y \in \mathcal{Y}} p(y|\boldsymbol{x})$, where $p(y|\boldsymbol{x})$ denotes the predicted probability for class $y$. Similarly, various confidence scoring functions can be adopted from previous work on OOD detection, such as MaxLogit (Hendrycks et al., 2022), Energy (Liu et al., 2020), Entropy (Chan et al., 2021), and Maximum Concept Matching (MCM) (Ming et al., 2022).

## 3 METHODOLOGY

### 3.1 LIMITATIONS OF THE BASELINES

Consistent with findings from previous work on misclassification detection research (Jaeger et al., 2022; Zhu et al., 2023), the simple MSP often outperforms more sophisticated OOD detection methods, as shown in Tab. 1. This observation suggests that advanced OOD detection methods frequently struggle to effectively capture misclassification errors in VLMs, underscoring the need to develop novel confidence-scoring functions tailored to this setting. Furthermore, the standard paradigm for zero-shot classification with VLM is mainly based on computing the cosine similarity between text and image embeddings (i.e., image-to-text similarity), as defined in Eq. (1). However, this approach often overlooks important characteristics of the image embedding space, such as image-to-image similarity. As demonstrated by Liang et al. (Liang et al., 2022), a modality gap exists within the representation space of VLMs; for example, the CLIP image and text embeddings reside in distinct regions of the joint embedding space, as illustrated in Fig. 1 (a). Consequently, relying solely on image-to-text similarity for zero-shot classification and misclassification detection may neglect critical information, potentially leading to suboptimal performance. For example, Fig. 1 (b) provides a concrete example using CLIP embeddings for the concepts 'dog' and 'seaplane'. In this case, the separation margin based on image-to-text similarity is only 0.13, whereas image-to-image similarity could yields a substantially larger margin of 0.29, indicating that the two concepts are more clearly distinguishable in the image embedding space.

This insight has practical implications. When an image-to-text prediction is incorrect – for instance, classifying an image of a dog as a 'seaplane' – the image-to-image similarity between the input image

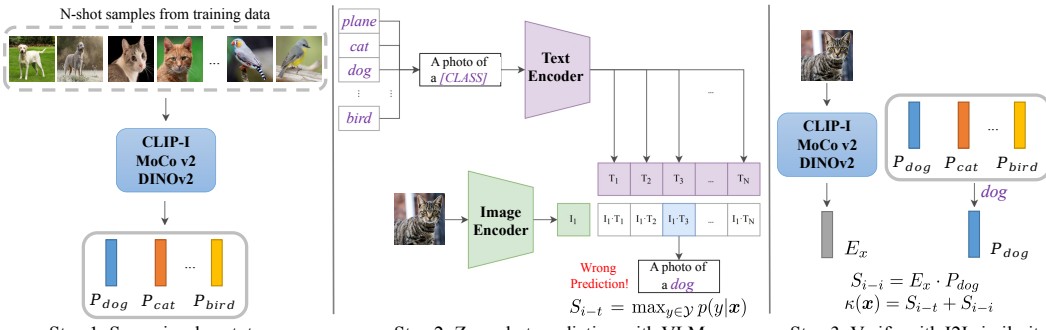

Figure 2: The proposed TrustVLM framework comprises three main steps. Initially, visual prototypes for each class are generated and stored using a pre-trained vision encoder. Subsequently, the VLMs perform zero-shot classification and yield an image-to-text similarity score, $S_{i-t}$. In the third step, the initial prediction is verified using image-to-image similarity, providing an additional confidence score, $S_{i-i}$. Finally, these two scores are combined to determine the overall prediction confidence.

and a visual prototype for 'seaplane' would likely be low, helping mitigate overconfidence. Conversely, for correct predictions (e.g., classifying a dog image as 'dog'), the image-to-image similarity with the corresponding prototype would generally be high, thereby reinforcing the prediction with greater confidence. Therefore, exploring image-to-image similarity is crucial for designing effective confidence-scoring functions to enhance misclassification detection performance in VLMs.

### 3.2 PROPOSED TRUSTVLM FRAMEWORK

Inspired by the modality gap phenomenon observed in VLMs and the enhanced distinguishability of certain concepts within the image embedding space, we propose TrustVLM. Our framework leverages information from the image embedding space to design the confidence-scoring function. In addition to the conventional confidence score derived from image-to-text similarity (calculated via Eq. (1)), TrustVLM incorporates a second score derived from image-to-image similarity and computed using an auxiliary vision encoder. These two scores are complementary, and their effective combination leads to more robust misclassification detection. The auxiliary vision encoder can be the original CLIP image encoder or other pre-trained vision models, such as MoCo v2 or DINOv2. The selection of a more powerful auxiliary vision encoder can further improve misclassification detection performance.

**Integrate Vision Encoder for Misclassification Detection.** Our *TrustVLM* framework operates in three main steps, as shown in Fig. 2. The first step involves generating and storing visual prototypes. Specifically, for each class $c$, embeddings are extracted from $N$-shot samples in the training data using a pre-trained vision encoder, $E$ (e.g., the CLIP image encoder, MoCo v2, or DINOv2). The prototype embedding for class $c$, $P_c$, is then computed by averaging these $N$ embeddings. These class prototypes $\{P_c\}$ are subsequently stored. In the second step, for a given input image $\boldsymbol{x}$, a zero-shot prediction $\hat{y}$ is obtained using the VLM, as defined in Eq. (1), where the prediction $\hat{y}$ could be either correct or wrong. Concurrently, an initial confidence score, $S_{i-t} = \max_{y \in \mathcal{Y}} p(y|\boldsymbol{x})$, is derived from the image-to-text similarity. The third step focuses on generating a complementary image-to-image confidence score. An embedding $E_x$ of the input image $\boldsymbol{x}$ is extracted using the same vision encoder $E$ employed in the first step. Since in the second step, the VLM believes $\boldsymbol{x}$ to be class $\hat{y}$, we calculate the cosine similarity between $E_x$ and $P_{\hat{y}}$ as the image-to-image similarity score $S_{i-i} = E_x \cdot P_{\hat{y}}$. This $S_{i-i}$ score is expected to be low if the prediction $\hat{y}$ is incorrect, as $E_x$ would be compared against an inappropriate prototype, thereby helping to mitigate overconfidence. Conversely, a correct prediction $\hat{y}$ should result in a high $S_{i-i}$, reinforcing the prediction's reliability. Finally, this verification mechanism yields the combined confidence score for input $\boldsymbol{x}$ is $\kappa(\boldsymbol{x}) = S_{i-t} + S_{i-i}$.

**Integrate Vision Encoder for Fine-grained Classification.** Visual prototypes and image-to-image similarity can also be utilized to enhance the fine-grained classification capabilities of VLMs. Given the visual prototypes $\{P_c\}$ for each class $c \in \mathcal{Y} = \{1, 2, \ldots, C\}$, the probability of predicting class

$\hat{y}$ for an input image $\boldsymbol{x}$ (with embedding $E_x$), based on image-to-image similarity, is computed as:

$$p(y = \hat{y}|\boldsymbol{x}) = \frac{\exp\left(\cos\left(E_x, P_{\hat{y}}\right)/\tau\right)}{\sum_{c=1}^{C} \exp\left(\cos\left(E_x, P_c\right)/\tau\right)}. \tag{3}$$

Combining Eq. (1) and Eq. (3), we get the ensemble prediction from both image-to-text and image-to-image similarity as:

$$p(y = \hat{y}|\boldsymbol{x}) = \frac{\exp\left(\cos\left(\mathbf{f}_x, \mathbf{t}_{\hat{y}}\right)/\tau\right)}{\sum_{c=1}^{C} \exp\left(\cos\left(\mathbf{f}_x, \mathbf{t}_c\right)/\tau\right)} + \frac{\exp\left(\cos\left(E_x, P_{\hat{y}}\right)/\tau\right)}{\sum_{c=1}^{C} \exp\left(\cos\left(E_x, P_c\right)/\tau\right)}. \tag{4}$$

For this variant, termed **_TrustVLM*_**, the confidence-scoring function for a given input $\boldsymbol{x}$ is $\kappa(\boldsymbol{x}) = \max_{y \in \mathcal{Y}} p(y|\boldsymbol{x}) + S_{i-i}$.

**Visual Prototypes with Fine-tuning.** Visual prototypes extracted from pre-trained vision encoders are typically fixed by default. In this section, we introduce **_TrustVLM*(F)_** to treat these visual prototypes as learnable parameters initialized with their pre-computed values. These parameters are subsequently fine-tuned using stochastic gradient descent. The rationale is that updating the visual prototypes can enhance affinity estimation, thereby enabling a more accurate calculation of cosine similarities between test and training images, as demonstrated by (Zhang et al., 2022). Specifically, we freeze the parameters of the VLMs and the vision encoder, while fine-tuning only the visual prototypes via a cross-entropy loss for 10 epochs with a learning rate of 0.001. We perform fine-tuning using the N-shot labeled samples from Step 1, where we compute predictions via Eq. (4) and optimize the cross-entropy loss between these predictions and the ground-truth labels. Finally, these learned prototypes replace the original fixed prototypes $\{P_c\}$ in Eq. (4). This fine-tuning step is lightweight. For example, on Flowers102 (Nilsback & Zisserman, 2008), it takes only 2 minutes on a single GeForce RTX 3090 GPU.

## 4 EXPERIMENTS

### 4.1 EXPERIMENTAL SETTING

**Dataset.** We evaluate our framework on a wide variety of 17 datasets. *Fine-grained Classification Datasets,* including 10 publicly available image classification datasets: Caltech101 (Fei-Fei et al., 2004), OxfordPets (Parkhi et al., 2012), StanfordCars (Krause et al., 2013), Flowers102 (Nilsback & Zisserman, 2008), Food101 (Bossard et al., 2014), FGVCAircraft (Maji et al., 2013), SUN397 (Xiao et al., 2010), DTD (Cimpoi et al., 2014), EuroSAT (Helber et al., 2019) and UCF101 (Soomro et al., 2012). These datasets constitute a comprehensive benchmark, which covers a diverse set of vision tasks including classification on generic objects, scenes, actions and fine-grained categories, as well as specialized downstream tasks such as recognizing textures and satellite imagery. *ImageNet and Its Variants,* including ImageNet (Deng et al., 2009), ImageNetV2 (Recht et al., 2019), ImageNet-Sketch (Wang et al., 2019), ImageNet-A (Hendrycks et al., 2021b), and ImageNet-R (Hendrycks et al., 2021a), with distribution shifts in image style, data domains, etc. We also evaluate our framework on CIFAR-10 and CIFAR-100 (Krizhevsky et al., 2009) to compare with ORCA (Nguyen et al., 2025).

**Implementation Details.** We utilize CLIP ViT-B/16 (Dosovitskiy et al., 2020) backbone to perform zero-shot prediction on the benchmarks and calculate the related performance metrics. We also compare with ORCA (Nguyen et al., 2025) on CLIP ResNet-101 (He et al., 2016) and ViT-B/32 following its setup. To demonstrate the generalization of the proposed framework to different VLMs, we further evaluate on CLIP ResNet-50 and SigLIP (Zhai et al., 2023) ViT-B/16. For the auxiliary vision encoder, we use both the original CLIP image encoder as well as other pre-trained models such as DINOv2 (Oquab et al., 2023) and MoCo v2 (Chen et al., 2020). We use 16-shot samples from the training data to calculate the prototypes by default and set the temperature $\tau$ to 0.01.

**Evaluation Metrics. AURC.** The area under the risk-coverage curve (AURC) (Geifman & El-Yaniv, 2017) depicts the error rate which is computed by using samples whose confidence is higher than some confidence thresholds. **AUROC.** The area under the receiver operating characteristic curve (AUROC) (Davis & Goadrich, 2006) depicts the relationship between true positive rate (TPR) and false positive rate (FPR). **FPR95.** The FPR at 95% TPR denotes the probability that a misclassified example is predicted as a correct one when the TPR is as high as 95%. **ACC.** Test accuracy (ACC) is also an important metric.

| | Flowers102 | | | | DTD | | | | Aircraft | | | | Pets | | | |
|---|---|---|---|---|---|---|---|---|---|---|---|---|---|---|---|---|
| | AURC↓ | AUROC↑ | FPR95↓ | ACC↑ | AURC↓ | AUROC↑ | FPR95↓ | ACC↑ | AURC↓ | AUROC↑ | FPR95↓ | ACC↑ | AURC↓ | AUROC↑ | FPR95↓ | ACC↑ |
| MaxLogit | 167.14 | 74.92 | 81.61 | 67.36 | 395.13 | 69.63 | 85.23 | 44.39 | 731.25 | 55.73 | 93.77 | 23.85 | 48.75 | 75.37 | 68.59 | 88.23 |
| Energy | 194.67 | 69.00 | 91.93 | 67.36 | 433.25 | 64.82 | 90.01 | 44.39 | 780.11 | 45.48 | 97.71 | 23.85 | 56.23 | 71.90 | 70.67 | 88.23 |
| Entropy | 117.27 | 84.88 | 63.48 | 67.36 | 319.02 | 79.03 | 77.05 | 44.39 | 575.71 | 73.12 | 83.96 | 23.85 | 21.91 | 89.31 | 53.35 | 88.23 |
| MCM | 153.63 | 78.22 | 71.02 | 67.36 | 333.43 | 78.16 | 78.45 | 44.39 | 583.28 | 72.58 | 81.65 | 23.85 | 44.95 | 77.03 | 64.90 | 88.23 |
| DOCTOR | 112.82 | 85.82 | 62.48 | 67.36 | 314.37 | 79.66 | 76.51 | 44.39 | 575.53 | 73.05 | 83.84 | 23.85 | 21.08 | 89.92 | 55.20 | 88.23 |
| MSP | 112.27 | 85.91 | 63.98 | 67.36 | 313.43 | 79.81 | 77.36 | 44.39 | 576.97 | 72.62 | 85.61 | 23.85 | 21.04 | 89.94 | 52.19 | 88.23 |
| TrustVLM-C | 101.42 | 88.69 | 54.91 | 67.36 | 302.18 | 82.52 | 67.27 | 44.39 | 563.77 | 75.20 | 81.51 | 23.85 | 20.93 | 89.89 | 51.73 | 88.23 |
| TrustVLM-M | 103.68 | 88.29 | 53.42 | 67.36 | 298.17 | 83.16 | 65.50 | 44.39 | 574.57 | 73.22 | 84.31 | 23.85 | 20.41 | 90.38 | 48.27 | 88.23 |
| TrustVLM-D | 77.30 | 95.05 | 30.06 | 67.36 | 268.71 | **88.55** | **44.10** | 44.39 | 562.02 | 75.62 | 83.21 | 23.85 | 20.69 | 90.05 | 50.81 | 88.23 |
| TrustVLM*-D | 0.52 | 95.96 | 13.04 | **99.07** | 124.15 | 78.39 | 72.14 | 71.57 | 554.12 | 75.36 | 83.05 | 24.60 | **20.05** | 90.30 | 49.07 | **88.28** |
| TrustVLM*(F)-D | **0.41** | **98.26** | **7.69** | 98.42 | **96.45** | 80.30 | 70.96 | **74.76** | **544.40** | **76.85** | **78.63** | 24.90 | **20.05** | 90.30 | 49.07 | **88.28** |

| | Caltech101 | | | | Cars | | | | EuroSAT | | | | UCF101 | | | |
|---|---|---|---|---|---|---|---|---|---|---|---|---|---|---|---|---|
| | AURC↓ | AUROC↑ | FPR95↓ | ACC↑ | AURC↓ | AUROC↑ | FPR95↓ | ACC↑ | AURC↓ | AUROC↑ | FPR95↓ | ACC↑ | AURC↓ | AUROC↑ | FPR95↓ | ACC↑ |
| MaxLogit | 53.72 | 52.43 | 94.55 | 93.31 | 259.81 | 61.94 | 89.58 | 65.61 | 522.56 | 59.42 | 87.79 | 42.10 | 258.47 | 61.85 | 91.41 | 65.21 |
| Energy | 59.26 | 48.81 | 96.36 | 93.31 | 304.90 | 54.85 | 93.89 | 65.61 | 568.57 | 53.44 | 88.30 | 42.10 | 299.84 | 55.38 | 94.83 | 65.21 |
| Entropy | 15.79 | 82.47 | 79.39 | 93.31 | 145.61 | 79.83 | 75.76 | 65.61 | 389.07 | 72.18 | 83.16 | 42.10 | 130.10 | 83.97 | 70.52 | 65.21 |
| MCM | 45.55 | 61.00 | 83.54 | 93.31 | 248.87 | 63.77 | 85.97 | 65.61 | 416.37 | 71.28 | 81.18 | 42.10 | 188.00 | 74.14 | 75.47 | 65.21 |
| DOCTOR | 12.89 | 86.06 | 76.36 | 93.31 | 138.00 | 81.50 | 73.66 | 65.61 | 368.77 | 74.43 | 81.69 | 42.10 | 123.41 | 85.67 | 65.81 | 65.21 |
| MSP | 12.23 | 86.99 | 67.88 | 93.31 | 136.27 | 81.95 | 72.25 | 65.61 | 355.66 | 76.39 | 80.65 | 42.10 | 122.44 | 85.98 | 64.89 | 65.21 |
| TrustVLM-C | 13.25 | 86.81 | 70.30 | 93.31 | 129.45 | **83.67** | 67.73 | 65.61 | 322.52 | 82.90 | 55.73 | 42.10 | 111.95 | 88.69 | 55.93 | 65.21 |
| TrustVLM-M | 10.81 | 88.97 | 64.24 | 93.31 | 134.25 | 82.53 | 72.29 | 65.61 | 320.12 | 83.03 | 56.32 | 42.10 | 114.60 | 87.93 | 58.81 | 65.21 |
| TrustVLM-D | 11.11 | 90.51 | 47.27 | 93.31 | 137.54 | 82.05 | 70.62 | 65.61 | 303.79 | 85.48 | 53.52 | 42.10 | 107.13 | **90.21** | 50.68 | 65.21 |
| TrustVLM*-D | 5.69 | 89.48 | **35.62** | 97.04 | 137.97 | 81.73 | 71.25 | 65.83 | 72.96 | 74.87 | 73.50 | 83.56 | 65.26 | 86.23 | 62.93 | 77.11 |
| TrustVLM*(F)-D | **2.61** | **93.29** | 35.71 | **97.16** | 132.54 | 82.58 | 68.70 | **66.02** | **54.35** | 77.31 | 72.48 | **85.69** | **55.81** | 87.26 | 66.06 | **78.35** |

| | Food101 | | | | SUN397 | | | | Average | | | |
|---|---|---|---|---|---|---|---|---|---|---|---|---|
| | AURC↓ | AUROC↑ | FPR95↓ | ACC↑ | AURC↓ | AUROC↑ | FPR95↓ | ACC↑ | AURC↓ | AUROC↑ | FPR95↓ | ACC↑ |
| MaxLogit | 67.81 | 75.81 | 78.13 | 83.66 | 278.17 | 62.77 | 90.46 | 62.57 | 278.28 | 64.99 | 86.11 | 63.63 |
| Energy | 76.05 | 72.33 | 84.69 | 83.66 | 304.50 | 58.34 | 93.47 | 62.57 | 307.74 | 59.44 | 90.19 | 63.63 |
| Entropy | 55.80 | 83.76 | 63.73 | 83.66 | 194.33 | 75.19 | 80.97 | 62.57 | 196.46 | 80.37 | 73.14 | 63.63 |
| MCM | 60.90 | 79.90 | 67.23 | 83.66 | 242.95 | 69.09 | 83.62 | 62.57 | 231.79 | 72.52 | 77.30 | 63.63 |
| DOCTOR | 54.99 | 84.39 | 61.11 | 83.66 | 184.41 | 77.35 | 77.37 | 62.57 | 190.63 | 81.78 | 71.40 | 63.63 |
| MSP | 54.80 | 84.51 | 59.73 | 83.66 | 182.39 | 77.90 | 76.42 | 62.57 | 188.75 | 82.20 | 70.10 | 63.63 |
| TrustVLM-C | 36.02 | 88.34 | 57.67 | 83.66 | 171.89 | 80.39 | 69.20 | 62.57 | 177.34 | 84.71 | 63.20 | 63.63 |
| TrustVLM-M | 40.92 | 86.65 | 59.01 | 83.66 | 172.98 | 79.83 | 71.63 | 62.57 | 179.05 | 84.40 | 63.38 | 63.63 |
| TrustVLM-D | 36.18 | 88.52 | **55.96** | 83.66 | 156.16 | **83.80** | **58.44** | 62.57 | 168.06 | **86.98** | 54.47 | 63.63 |
| TrustVLM*-D | 35.01 | 88.49 | 57.47 | 84.10 | 104.25 | 82.31 | 72.82 | **72.18** | 112.00 | 84.31 | 59.09 | 76.33 |
| TrustVLM*(F)-D | **34.16** | **88.79** | 56.19 | **84.15** | **103.05** | 83.43 | 72.68 | 71.33 | **104.38** | 85.84 | 57.82 | **76.91** |

Table 1: Misclassification detection performance on fine-grained classification datasets with CLIP ViT-B/16, where -C, -M, and -D are with CLIP-I, MoCo v2, and DINOv2 as auxiliary vision encoders. AURC is multiplied by $10^3$ following previous work Zhu et al. (2023).

**Baselines.** We compare our method against well-established confidence-scoring functions, including MaxLogit (Hendrycks et al., 2022), Energy (Liu et al., 2020), Entropy (Chan et al., 2021), MCM (Ming et al., 2022), MSP (Hendrycks & Gimpel, 2017), and DOCTOR (Granese et al., 2021), where DOCTOR fully exploits all available information contained in the soft-probabilities of the predictions to estimate the confidence. We also compare with the most recent concept-based method ORCA (Nguyen et al., 2025).

## 4.2 RESULTS

**MisD Results on Fine-grained Classification Datasets.** As presented in Tab. 1, the simple MSP baseline consistently surpasses prominent OOD detection methods in MisD, including MaxLogit, Energy, and MCM. This indicates that current OOD detection techniques are limited in capturing misclassification errors effectively, highlighting a promising direction for future research: the development of confidence estimation methods that integrate OOD detection and MisD within a unified framework. Incorporating an image-to-image similarity confidence score in TrustVLM significantly enhances MisD performance, irrespective of the vision encoder employed (e.g., CLIP-I, MoCo v2, or DINOv2), thereby demonstrating the proposed framework's versatility. Notably, TrustVLM-D utilizing the DINOv2 encoder yields the best overall performance, achieving average improvements of 20.69% in AURC, 4.78% in AUROC, and 15.63% in FPR95 relative to the strongest baseline.

When visual prototypes are employed for zero-shot classification (TrustVLM*-D), the accuracy improves substantially by an average of 12.7%. This enhanced accuracy reduces the overall risk (error rate) across all coverage levels; consequently, the Area Under the Risk-Coverage curve (AURC) also decreases markedly, by an average of 56.06% compared to TrustVLM-D. However, its AUROC and FPR95 metrics are marginally lower than those of TrustVLM-D. A potential explanation for

| | ImageNet-A | | | | ImageNet-V2 | | | | ImageNet-R | | | |
|---|---|---|---|---|---|---|---|---|---|---|---|---|
| | AURC↓ | AUROC↑ | FPR95↓ | ACC↑ | AURC↓ | AUROC↑ | FPR95↓ | ACC↑ | AURC↓ | AUROC↑ | FPR95↓ | ACC↑ |
| MaxLogit | 387.23 | 64.16 | 86.59 | 50.08 | 254.37 | 68.58 | 85.22 | 61.20 | 138.25 | 72.98 | 79.48 | 74.19 |
| Energy | 421.78 | 59.15 | 90.00 | 50.08 | 278.89 | 64.64 | 89.26 | 61.20 | 160.84 | 67.39 | 86.55 | 74.19 |
| Entropy | 298.23 | 75.46 | 77.47 | 50.08 | 189.58 | 77.57 | 79.90 | 61.20 | 79.64 | 85.80 | 63.13 | 74.19 |
| MCM | 341.67 | 71.99 | 78.27 | 50.08 | 236.32 | 71.62 | 80.38 | 61.20 | 109.67 | 80.49 | 64.73 | 74.19 |
| DOCTOR | 290.78 | 76.67 | 76.93 | 50.08 | 181.67 | 79.32 | 73.67 | 61.20 | 75.33 | 87.14 | 60.81 | 74.19 |
| MSP | 290.46 | 76.70 | 77.06 | 50.08 | 180.32 | 79.72 | 71.82 | 61.20 | 74.36 | 87.50 | 59.66 | 74.19 |
| TrustVLM-D | 274.46 | **79.17** | **70.27** | 50.08 | 176.82 | **81.27** | **66.02** | 61.20 | 72.10 | **88.03** | **57.70** | 74.19 |
| TrustVLM*-D | 266.48 | 75.94 | 77.67 | **53.02** | 172.65 | 77.93 | 74.92 | 63.92 | 71.72 | 87.50 | 60.03 | 74.62 |
| TrustVLM*(F)-D | **264.36** | 77.09 | 76.90 | 52.61 | 176.13 | 77.71 | 72.98 | **63.98** | **71.15** | 87.42 | 60.37 | **74.83** |
| | ImageNet-Sketch | | | | ImageNet | | | | Average | | | |
| | AURC↓ | AUROC↑ | FPR95↓ | ACC↑ | AURC↓ | AUROC↑ | FPR95↓ | ACC↑ | AURC↓ | AUROC↑ | FPR95↓ | ACC↑ |
| MaxLogit | 422.41 | 65.28 | 87.25 | 45.67 | 228.56 | 65.95 | 85.80 | 66.68 | 286.16 | 67.39 | 84.87 | 59.56 |
| Energy | 472.36 | 58.03 | 92.25 | 45.67 | 253.77 | 61.61 | 89.95 | 66.68 | 317.53 | 62.16 | 89.60 | 59.56 |
| Entropy | 313.83 | 78.37 | 74.27 | 45.67 | 149.69 | 78.26 | 78.01 | 66.68 | 206.19 | 79.09 | 74.56 | 59.56 |
| MCM | 400.13 | 70.10 | 76.52 | 45.67 | 206.72 | 69.73 | 81.27 | 66.68 | 258.90 | 72.79 | 76.23 | 59.56 |
| DOCTOR | 300.85 | 80.49 | 70.75 | 45.67 | 140.70 | 80.48 | 74.46 | 66.68 | 197.87 | 80.82 | 71.32 | 59.56 |
| MSP | 299.57 | 80.74 | 70.51 | 45.67 | 138.99 | 81.00 | 72.74 | 66.68 | 196.74 | 81.13 | 70.36 | 59.56 |
| TrustVLM-D | 280.72 | **84.35** | **58.99** | 45.67 | 132.10 | **83.38** | **64.04** | 66.68 | 187.24 | **83.24** | **63.40** | 59.56 |
| TrustVLM*-D | 264.51 | 74.74 | 84.77 | 53.13 | **126.31** | 78.82 | 77.43 | 70.58 | 180.33 | 78.99 | 74.96 | 63.05 |
| TrustVLM*(F)-D | **243.44** | 73.80 | 76.90 | **58.20** | 129.38 | 77.33 | 75.28 | **71.75** | **176.89** | 78.67 | 72.49 | **64.27** |

Table 2: Misclassification detection performance on ImageNet and its variants with CLIP ViT-B/16.

this is that as model accuracy improves, the confidence scores for many correct predictions might increase significantly. Nevertheless, some correctly classified but inherently "difficult" instances may still receive lower confidence scores. If the confidence scores of the few remaining misclassifications become more similar to these "hard but correct" instances, the overlap between the confidence distributions of misclassifications and correct classifications increases. This makes it harder for the detector to find a good threshold. Finally, fine-tuning on visual prototypes (TrustVLM*(F)-D) further enhances performance across all evaluated metrics compared to TrustVLM*-D.

**MisD Results on ImageNet and Its Variants.**
Tab. 2 presents results from large-scale experiments conducted on ImageNet and its variants. As the variant datasets solely provide test splits, $N$ samples per class were selected from these test sets to compute prototypes, with the remaining samples utilized for evaluation. Consistent with the findings in Tab. 1, the simple MSP baseline consistently outperforms strong OOD detection methods. Our TrustVLM-D with DINOv2 encoder demonstrates superior overall performance in most cases, achieving average improvements of 9.5% in AURC, 2.11% in AUROC, and 6.96% in FPR95 relative to the strongest baseline. While TrustVLM*-D and

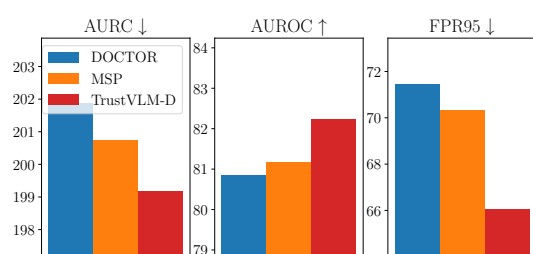

Figure 3: MisD performance on ImageNet and its variants using CLIP ViT-B/16, under distribution shifts defined by prototypes computed exclusively on ImageNet and deployed directly to variant datasets.

TrustVLM*(F)-D yield significant improvements in ACC and AURC, they adversely affect AUROC and FPR95 on these large-scale benchmarks. To further evaluate the robustness of our proposed method to distribution shifts, prototypes were computed using only N samples per class from the ImageNet training split. These prototypes were then applied directly to the ImageNet variants, thereby obviating the need to compute variant-specific prototypes. As illustrated in Fig. 3, our method demonstrates the overall leading performance of all metrics evaluated in this challenging scenario. Detailed results are provided in Tab. 10.

**Comparison with Concept-based Method.** We further compare our method against the most recent ORCA (Nguyen et al., 2025) with CLIP ResNet-101 and ViT-B/32 on CIFAR-10, CIFAR-100, and EuroSAT. ORCA leverages human-level concepts to detect when and interpret why a model fails. As shown in Tab. 3, while ORCA generally outperforms the MSP baseline in most cases, our proposed method demonstrates a significant performance margin over ORCA. Specifically, with the CLIP

| | | CIFAR-10 | | | CIFAR-100 | | | EuroSAT | | | Average | | |
|---|---|---|---|---|---|---|---|---|---|---|---|---|---|
| | | AUROC↑ | FPR95↓ | ACC↑ | AUROC↑ | FPR95↓ | ACC↑ | AUROC↑ | FPR95↓ | ACC↑ | AUROC↑ | FPR95↓ | ACC↑ |
| ResNet-101 | MSP | 85.98 | 62.98 | 78.01 | 80.72 | 73.40 | 48.50 | 61.73 | 88.98 | 30.30 | 76.14 | 75.12 | 52.27 |
| | ORCA | 85.93 | 62.68 | 80.60 | 80.46 | 72.38 | 53.11 | 69.01 | 86.43 | 34.76 | 78.47 | 73.83 | 56.16 |
| | TrustVLM-D | **94.76** | **26.82** | 78.01 | **90.03** | **39.46** | 48.50 | **74.03** | **66.69** | 30.30 | **86.27** | **44.32** | 52.27 |
| | TrustVLM*-D | 90.97 | 41.88 | **95.63** | 82.27 | 65.52 | **79.35** | 72.08 | 72.15 | **81.78** | 81.77 | 59.85 | **85.59** |
| ViT-B/32 | MSP | 88.92 | 58.66 | 88.92 | 81.15 | 71.09 | 58.42 | 76.42 | 80.24 | 41.11 | 82.16 | 70.00 | 62.82 |
| | ORCA | 89.00 | 52.70 | 90.00 | 83.40 | 67.00 | 66.50 | 77.55 | 71.29 | 50.00 | 83.32 | 63.66 | 68.83 |
| | TrustVLM-D | **94.94** | 31.85 | 88.92 | **89.13** | **45.63** | 58.42 | **78.88** | **69.36** | 41.11 | **87.65** | **48.95** | 62.82 |
| | TrustVLM*-D | 94.06 | **29.58** | **95.91** | 84.95 | 62.99 | **79.98** | 74.70 | 72.77 | **82.81** | 84.57 | 55.11 | **86.23** |

Table 3: MisD performance compared with ORCA with CLIP ResNet-101 and ViT-B/32.

| Method | Aircraft | Caltech101 | Cars | DTD | EuroSAT | Flowers102 | Food101 | Pets | SUN397 | UCF101 | Average |
|---|---|---|---|---|---|---|---|---|---|---|---|
| CLIP-RN50 | 15.54 | 85.88 | 55.74 | 40.37 | 23.70 | 61.75 | 73.95 | 83.65 | 58.81 | 58.74 | 55.81 |
| CoOp (Zhou et al., 2022) | 22.20 | 87.70 | 61.30 | 52.20 | 63.20 | 81.00 | 76.30 | 86.20 | 63.40 | 67.00 | 66.05 |
| Tip-Adapter (Zhang et al., 2022) | 23.70 | 88.80 | 63.90 | 54.70 | 72.50 | 83.20 | 76.70 | 86.40 | 66.70 | 72.10 | 68.87 |
| CuPL (Pratt et al., 2023) | 19.59 | 89.29 | 57.28 | 48.64 | 38.38 | 65.44 | 76.94 | 84.84 | 62.55 | 58.97 | 60.19 |
| TPT (Shu et al., 2022) | 17.58 | 87.02 | 58.46 | 40.84 | 28.33 | 62.69 | 74.88 | 84.49 | 61.46 | 60.82 | 57.66 |
| DMN (Zhang et al., 2024) | 20.22 | 89.09 | 58.36 | 50.53 | 44.94 | 68.33 | 74.69 | 86.29 | 63.70 | 64.02 | 62.02 |
| TDA (Karmanov et al., 2024) | 17.61 | 89.70 | 57.78 | 43.74 | 42.11 | 68.74 | **77.75** | 86.18 | 62.53 | 64.18 | 61.03 |
| ECALP (Li et al., 2025) | 21.12 | 89.94 | 60.56 | 54.49 | 49.09 | 69.39 | 76.97 | 88.20 | 64.97 | 66.67 | 64.14 |
| TrustVLM*-C | **29.34** | 89.98 | **66.09** | 60.34 | 75.11 | 90.62 | 74.53 | 85.94 | 66.30 | 72.09 | 71.03 |
| TrustVLM*-M | 19.89 | 93.27 | 57.52 | 65.43 | **86.85** | 87.25 | 74.85 | **90.22** | 67.16 | 73.28 | 71.57 |
| TrustVLM*-D | 20.25 | 96.43 | 56.08 | 71.34 | 82.43 | **99.11** | 75.23 | 83.65 | **71.45** | 75.34 | 73.13 |
| TrustVLM*(F)-D | 27.27 | **96.59** | 56.19 | **74.70** | 85.43 | 98.50 | 75.17 | 83.76 | 70.71 | **76.90** | **74.52** |

Table 4: Classification results on fine-grained datasets with CLIP ResNet-50.

ResNet-101 backbone, our approach achieves maximum improvements over ORCA of $9.57\%$ in AUROC, $35.86\%$ in FPR95, and $47.02\%$ in ACC. When employing the CLIP ViT-B/32 backbone, these respective maximum improvements are $5.94\%$, $21.37\%$, and $32.81\%$.

**Improved Classification Results on Fine-grained Datasets.** Visual prototypes and image-to-image similarity can also enhance the prediction accuracy of VLMs. In Tab. 4, we compare our method against various zero-shot, few-shot, and test-time adaptation baselines using CLIP ResNet-50. Notably, without requiring any training phase, our method achieves the best overall performance, yielding an average accuracy improvement of $4.36\%$ relative to these baselines. Furthermore, our approach is compatible with diverse vision encoders, including CLIP-I, MoCo v2, and DINOv2, and demonstrates robust performance across all of them. Subsequent fine-tuning of the visual prototypes, as implemented in TrustVLM*(F)-D, further improves accuracy by an additional $1.29\%$ compared to TrustVLM*-D.

## 4.3 Ablation Studies

**Different Architectures and VLMs.** To demonstrate the versatility of the proposed framework, we evaluated its performance with different architectures and VLMs. Specifically, we replaced the CLIP ViT-B/16 backbone in Tab. 1 with CLIP ResNet-50 and SigLIP ViT-B/16, reporting average performance metrics across 10 fine-grained datasets in Tab. 5. Consistent with the observations in Tab. 1, our method demonstrates robust compatibility across these varied architectures and VLMs, significantly surpassing the baseline methods in both configurations. More detailed results, including performance on ImageNet and its variants, are provided in the Appendix.

| | CLIP ResNet-50 | | | | SigLIP ViT-B/32 | | | |
|---|---|---|---|---|---|---|---|---|
| | AURC↓ | AUROC↑ | FPR95↓ | ACC↑ | AURC↓ | AUROC↑ | FPR95↓ | ACC↑ |
| DOCTOR | 259.56 | 80.55 | 73.83 | 55.82 | 149.27 | 85.69 | 58.63 | 70.69 |
| MSP | 259.76 | 80.63 | 73.11 | 55.82 | 149.13 | 85.86 | 57.31 | 70.69 |
| TrustVLM-D | 226.95 | **87.31** | **53.98** | 55.82 | 129.84 | **90.35** | **44.49** | 70.69 |
| TrustVLM*-D | **135.81** | 83.51 | 59.97 | **73.13** | **87.35** | 87.47 | 53.97 | **79.99** |

Table 5: Ablation on different architectures and VLMs. The average results on fine-grained classification datasets are reported.

| i-t | i-i | AURC↓ | AUROC↑ | FPR95↓ |
|---|---|---|---|---|
| ✓ | | 188.75 | 82.20 | 70.10 |
| | ✓ | 214.14 | 77.15 | 62.26 |
| ✓ | ✓ | **168.06** | **86.98** | **54.47** |

Table 6: Ablation on each component.

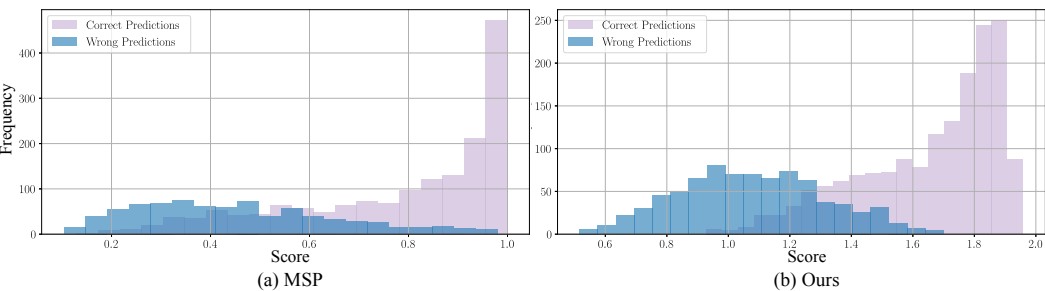

Figure 4: Score distribution for correct and wrong predictions. Our TrustVLM achieves better separation between the score distributions of correct and wrong predictions, leading to improved performance in misclassification detection.

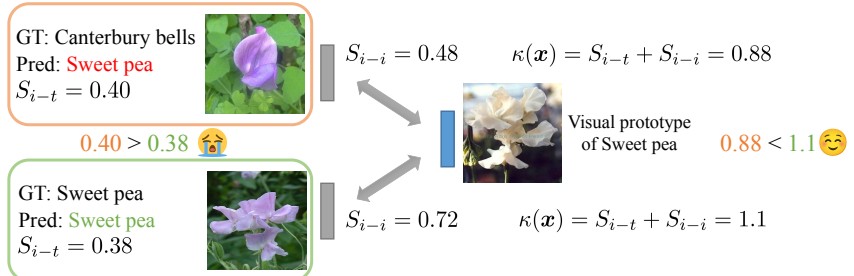

Figure 5: Illustration of TrustVLM's mechanism. Initially, the incorrect prediction receives a higher confidence score $S_{i-t}$ than the correct one, indicating overconfidence. By performing verification in the image embedding space using $S_{i-i}$, this overconfidence is mitigated. As a result, the final confidence score $\kappa(\boldsymbol{x})$ is significantly higher for the correct prediction than for the incorrect one.

**Ablation on Each Component.** We conducted comprehensive ablation studies to evaluate the contribution of each proposed module, as detailed in Tab. 6. We denote 'i-t' as a baseline relying solely on image-to-text similarity (akin to MSP), and 'i-i' as the confidence score derived from our proposed image-to-image similarity module, which utilizes a vision encoder. The results indicate that employing either the 'i-t' or 'i-i' component alone yields suboptimal performance. These findings underscore the complementary nature of the two components, with the best results obtained when they are used in combination.

**Visualization.** We visualized the confidence score distributions for correct and incorrect predictions on the Flowers102 dataset in Fig. 4. The baseline MSP exhibits a poorer separation in confidence scores between correctly classified and misclassified samples. In contrast, our solution assigns higher confidence scores to correct predictions and lower scores to incorrect ones, leading to more distinct distributions and, thereby, improved misclassification detection. Fig. 5 illustrates TrustVLM's mechanism for mitigating overconfidence, with more examples provided in Fig. 7.

## 5 CONCLUSION

In this work, we tackle the critical issue of misclassifications in VLMs, which hinders their reliable use, especially in safety-sensitive domains. We introduce TrustVLM, a novel training-free framework that substantially improves misclassification detection. TrustVLM uniquely leverages the commonly overlooked image embedding space by incorporating image-to-image similarity with an auxiliary vision encoder to derive a more discerning confidence score. The auxiliary vision encoder can also help VLMs make better predictions on fine-grained datasets and can be fine-tuned to achieve better performance. Our rigorous evaluations across 17 datasets, 4 architectures, and 2 VLMs demonstrated TrustVLM's state-of-the-art performance. These findings highlight the considerable benefits of our approach in identifying confident, yet incorrect, VLM predictions. By enhancing the ability to determine when VLM outputs are trustworthy, TrustVLM contributes to the safer and more dependable deployment of these powerful models in real-world scenarios.

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

## A    THE USE OF LARGE LANGUAGE MODELS

In preparing this manuscript, large language models (LLMs) were employed in a limited capacity as general-purpose writing assistants. Specifically, they were used to improve clarity of expression, refine grammar and sentence structure, and correct typographical or spelling errors. The models did not contribute to the generation of research ideas, the design of experiments, the analysis or interpretation of results, or the formulation of the scientific claims presented in this work.

## B    BROADER IMPACT, LIMITATIONS, AND FUTURE WORK

**Broader Impact.** The development of TrustVLM offers significant positive societal impacts by directly addressing the critical need for more reliable and trustworthy Vision-Language Models (VLMs). As VLMs become increasingly integrated into real-world applications, particularly in safety-critical domains such as autonomous driving, medical diagnostics, and public safety surveillance, the ability to accurately discern when a model's prediction can be trusted is paramount. Erroneous and overconfident predictions in these areas can lead to severe adverse consequences. TrustVLM contributes to mitigating such risks by providing a robust framework for misclassification detection, thereby enhancing the safety of VLM-powered systems.

**Limitations.** While TrustVLM demonstrates strong performance across diverse benchmarks, several limitations remain. First, our method assumes access to clean class-level visual prototypes, which may not always be feasible in noisy or open-world settings. Second, our current work primarily focuses on zero-shot classification. While the core principles may be adaptable, the direct applicability and performance of TrustVLM on other VLM tasks, such as visual question answering or image captioning, have not yet been extensively evaluated. Third, the approach relies on a fixed auxiliary vision encoder and does not account for scenarios where the underlying data distribution evolves over time, such as in continual learning or streaming environments.

**Future Work.** Building upon the promising results of TrustVLM, several avenues for future research warrant exploration. A key direction is the extension of the TrustVLM framework to a broader range of multimodal tasks beyond zero-shot classification, including visual question answering, image retrieval, and image captioning, to assess its generalizability and adapt its mechanisms where necessary. Besides, incorporating human-in-the-loop feedback for refining confidence scores may further improve VLM reliability in complex, real-world deployments. Finally, it is also an interesting direction to explore the use of LLMs for correcting the model's prediction and identifying the true class, as demonstrated in recent work (Huang et al., 2023).

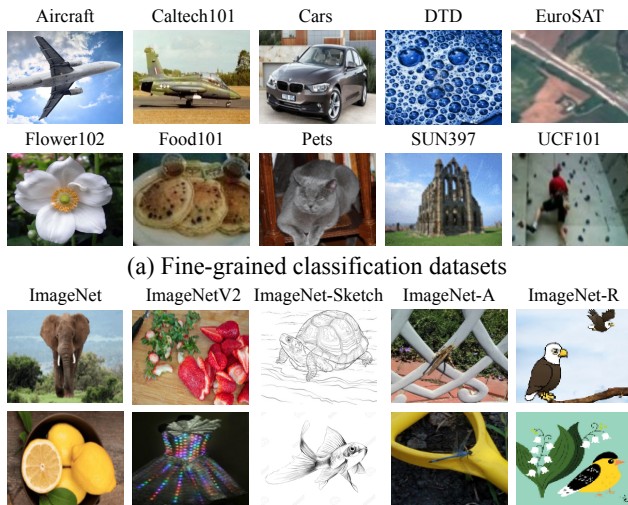

(a) Fine-grained classification datasets

(b) ImageNet and its variants

Figure 6: Representative examples from each dataset used in this work.

## C  RELATED WORK

**Misclassification Detection.** The primary goal of MisD is to distinguish misclassified samples from correctly classified ones, often by evaluating the reliability of the prediction. Early approaches used simple baselines such as Maximum Softmax Probability (MSP) (Hendrycks & Gimpel, 2017), although these were limited by model overconfidence. TrustScore (Jiang et al., 2018) estimates the reliability of predictions based on distances to training samples in the feature space, though it can struggle with high-dimensional or domain-shifted data. DOCTOR (Granese et al., 2021) introduces a simple rule-based rejection mechanism for black-box models without retraining, yet its performance depends on carefully tuned thresholds. Another direction involves directly learning confidence scores or training auxiliary components to predict prediction failure, such as learning the true class probability (Corbière et al., 2019) or adding dedicated confidence branches (DeVries & Taylor, 2018). OpenMix (Zhu et al., 2023) enhances robustness by generating synthetic outliers during training, improving calibration on misclassified samples. Recent literature (Nguyen et al., 2025) has also explored MisD with VLMs by leveraging human-level concepts to detect when and interpret why a model fails.

**Out-of-distribution Detection** shares a similar objective with MisD but addresses fundamentally distinct challenges. OOD detection aims to identify test samples that exhibit semantic shifts without compromising in-distribution (ID) classification accuracy, which can be broadly categorized into post hoc methods and training-time regularization. *Post hoc* methods design OOD scores based on the classification outputs of neural networks, offering the advantage of ease of use without modifying the training procedure or objective (Hendrycks & Gimpel, 2017; Hendrycks et al., 2022; Liu et al., 2020). *Training-time regularization* methods address prediction overconfidence by imposing a constant vector norm on the logits during training (Wei et al., 2022) or using external OOD samples from other datasets during training to improve discrimination between ID and OOD samples (Hendrycks et al., 2019; Nejjar et al., 2024). Recently, some works (Ming et al., 2022; Jiang et al., 2024; Wang et al., 2023) have also explored OOD detection via use of VLMs. However, methods optimized for OOD detection often underperform on MisD (Jaeger et al., 2022; Zhu et al., 2023), underscoring the need for specialized MisD approaches.

## D  MORE DETAILS ON THE DATASETS

We mainly evaluate our framework on 15 datasets from *Fine-grained Classification Datasets* and *ImageNet and Its Variants*. The fine-grained classification datasets include 10 publicly available image classification datasets, covering species of plants or animals (Flowers102 (Nilsback & Zisserman, 2008), OxfordPets (Parkhi et al., 2012)), scenes (SUN397 (Xiao et al., 2010)), textures (DTD (Cimpoi

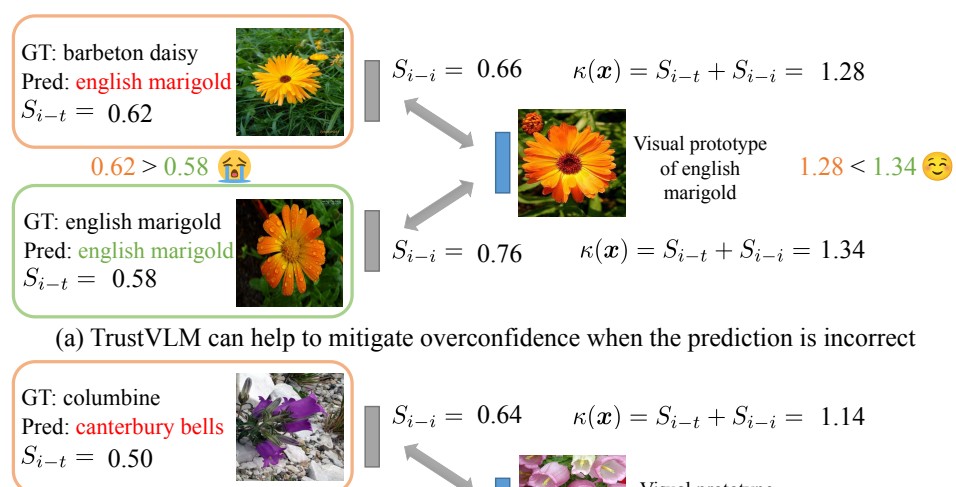

(a) TrustVLM can help to mitigate overconfidence when the prediction is incorrect

(b) TrustVLM can reinforce the prediction's reliability when the prediction is correct

Figure 7: More illustration on TrustVLM's mechanism.

et al., 2014)), food (Food101 (Bossard et al., 2014)), transportation(StanfordCars (Krause et al., 2013), FGVCAircraft (Maji et al., 2013)), human actions (UCF101 (Soomro et al., 2012)), satellite images (EuroSAT (Helber et al., 2019)), and general objects (Caltech101 (Fei-Fei et al., 2004)). The ImageNet and Its Variants features the original ImageNet (Deng et al., 2009), along with several key variants: ImageNetV2 (Recht et al., 2019), an independent test set with natural images from a different source; ImageNet-Sketch (Wang et al., 2019), comprising black and white sketches; ImageNet-A (Hendrycks et al., 2021b), a challenging test set of 'natural adversarial examples' often misclassified by standard ResNet-50 models (He et al., 2016); and ImageNet-R (Hendrycks et al., 2021a), featuring artistic renditions of ImageNet categories. These variants collectively introduce diverse distribution shifts in image style, data domains, and other factors. Fig. 6 illustrates representative examples from these datasets.

## E  FURTHER EXPERIMENTAL RESULTS

**More illustration on TrustVLM's mechanism.** Fig. 7 demonstrates more examples on how TrustVLM works. The $S_{i-i}$ score is expected to be low if the prediction $\hat{y}$ is incorrect, as the embedding $E_x$ would be compared against an inappropriate prototype, thereby helping to mitigate overconfidence. Conversely, a correct prediction $\hat{y}$ should result in a high $S_{i-i}$, reinforcing the prediction's reliability.

**Influence of N.** We investigated the effect of $N$, the number of samples per class used for computing prototypes, on AUROC performance. As illustrated in Fig. 8, performance steadily improves with an increasing number of samples per class. Notably, using only a single sample per class ($N = 1$) already achieves results superior to the baseline. The performance improvement tends to saturate when $N$ exceeds 4.

**Robustness to the selection of N-shot samples.** To evaluate the robustness of our method with respect to the selection of N-shot samples, we randomly sample three different sets of only one image per class (N=1) from the training data. As shown in Tab. 7, the performance remains consistent across different sample sets. This suggests that our framework is not overly sensitive to the specific choice of labeled examples, and can generalize well even when using a single randomly chosen image per class. For further evaluation, we randomly sample three different sets of 4 images per class from the training

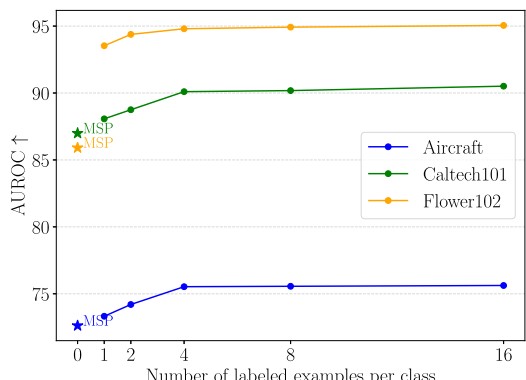

Figure 8: Influence of $N$ in prototypes.

|  | AURC↓ | AUROC↑ | FPR95↓ |
|---|---|---|---|
| Set 1 | 82.40 | 93.53 | 38.39 |
| Set 2 | 80.83 | 94.02 | 33.91 |
| Set 3 | 80.96 | 93.97 | 34.66 |

Table 7: Robustness to the selection of N-shot samples (TrustVLM-D on Flowers102 with N=1).

|  | AURC↓ | AUROC↑ | FPR95↓ |
|---|---|---|---|
| Set 1 | 78.15 | 94.80 | 29.69 |
| Set 2 | 78.47 | 94.74 | 29.07 |
| Set 3 | 78.18 | 94.81 | 29.81 |

Table 8: Robustness to the selection of N-shot samples (TrustVLM-D on Flowers102 with N=4).

|  | AURC↓ | AUROC↑ | FPR95↓ |
|---|---|---|---|
| MSP | 122.44 | 85.98 | 64.89 |
| TrustVLM-D (SD3) | 119.90 | 86.96 | 62.31 |
| TrustVLM-D | 107.13 | 90.21 | 50.68 |

Table 9: Robustness under privacy-sensitive settings. We use Stable Diffusion 3 (SD3) Medium to generate 16 images per class for UCF101 to calculate visual prototypes.

data and report performance for each. As shown in Tab. 8, the results remain consistently strong and stable, confirming that our method is not overly sensitive to the specific choice of representative samples. This also shows that increasing the number of labeled examples per class is an effective way to mitigate the effect of outlier or noisy samples in the embedding space.

**Robustness under privacy-sensitive settings.** Our framework is designed to be highly data-efficient. As shown in Fig. 8, using just a single labeled sample per class is sufficient for our method to outperform the baseline, demonstrating its effectiveness even in low-data regimes. To address the scenario without labeled data, we explored using a generative model as a substitute. We conducted a new experiment using Stable Diffusion 3 (SD3) Medium (Rombach et al., 2022) to generate 16 images per class for UCF101 (e.g., with the prompt "a photo of a person doing [action name]"). We call this variant TrustVLM-D (SD3). As shown in Tab. 9, TrustVLM-D (SD3) outperforms the strong MSP baseline, even though a distribution gap between generated images and real images prevents it from matching the performance of using real data. This result validates the feasibility of using generated data as an alternative when labeled examples are unavailable. Additionally, for privacy-sensitive settings, we propose a practical deployment strategy: clients can locally generate prototypes using their private labeled data, and only share the prototype embeddings (not the raw data) with the system. Since our framework operates on these prototypes, access to raw sensitive data is not required, making it a privacy-conscious solution suitable for high-risk domains.

| | ImageNet-A | | | | ImageNet-V2 | | | | ImageNet-R | | | |
|---|---|---|---|---|---|---|---|---|---|---|---|---|
| | AURC↓ | AUROC↑ | FPR95↓ | ACC↑ | AURC↓ | AUROC↑ | FPR95↓ | ACC↑ | AURC↓ | AUROC↑ | FPR95↓ | ACC↑ |
| DOCTOR | 316.50 | 75.98 | 77.76 | 47.85 | 180.06 | 80.09 | 73.38 | 60.88 | 76.39 | 87.12 | 60.54 | 73.98 |
| MSP | **315.99** | **76.07** | 77.76 | 47.85 | 178.82 | 80.45 | 71.26 | 60.88 | 75.47 | 87.46 | 59.43 | 73.98 |
| TrustVLM-D | 324.07 | 75.87 | **77.20** | 47.85 | **174.23** | **82.18** | **64.82** | 60.88 | **74.94** | **87.69** | **58.09** | 73.98 |
| | ImageNet-Sketch | | | | ImageNet | | | | Average | | | |
| | AURC↓ | AUROC↑ | FPR95↓ | ACC↑ | AURC↓ | AUROC↑ | FPR95↓ | ACC↑ | AURC↓ | AUROC↑ | FPR95↓ | ACC↑ |
| DOCTOR | 296.20 | 80.54 | 70.94 | 46.09 | 140.21 | 80.53 | 74.58 | 66.71 | 201.87 | 80.85 | 71.44 | 59.10 |
| MSP | 294.85 | 80.82 | 70.29 | 46.09 | 138.52 | 81.04 | 72.85 | 66.71 | 200.73 | 81.17 | 70.32 | 59.10 |
| TrustVLM-D | **289.52** | **82.40** | **65.59** | 46.09 | **133.17** | **83.08** | **64.47** | 66.71 | **199.19** | **82.24** | **66.03** | 59.10 |

Table 10: MisD performance on ImageNet and its variants using CLIP ViT-B/16, under distribution shifts defined by prototypes computed exclusively on ImageNet and deployed directly to variant datasets.

| | Flowers102 | | | | DTD | | | | Aircraft | | | | Pets | | | |
|---|---|---|---|---|---|---|---|---|---|---|---|---|---|---|---|---|
| | AURC↓ | AUROC↑ | FPR95↓ | ACC↑ | AURC↓ | AUROC↑ | FPR95↓ | ACC↑ | AURC↓ | AUROC↑ | FPR95↓ | ACC↑ | AURC↓ | AUROC↑ | FPR95↓ | ACC↑ |
| DOCTOR | 26.03 | 93.13 | 38.75 | 83.76 | 183.86 | 79.14 | 72.67 | 61.29 | 513.33 | 78.08 | 80.74 | 27.42 | 7.65 | 95.55 | 30.65 | 91.55 |
| MSP | 25.14 | 93.63 | 34.75 | 83.76 | 181.52 | 79.66 | 72.52 | 61.29 | 514.42 | 77.78 | 76.98 | 27.42 | 7.65 | 95.54 | 28.71 | 91.55 |
| TrustVLM-D | 16.62 | 98.42 | 9.50 | 83.76 | 144.44 | 87.64 | 49.01 | 61.29 | 503.98 | 79.56 | 75.83 | 27.42 | 7.66 | 95.58 | 28.71 | 91.55 |
| TrustVLM*-D | 0.13 | 97.82 | 15.38 | 99.47 | 98.27 | 82.72 | 71.92 | 72.64 | 496.11 | 78.79 | 79.14 | 28.23 | 7.55 | 95.56 | 29.97 | 91.63 |
| | Caltech101 | | | | Cars | | | | EuroSAT | | | | UCF101 | | | |
| | AURC↓ | AUROC↑ | FPR95↓ | ACC↑ | AURC↓ | AUROC↑ | FPR95↓ | ACC↑ | AURC↓ | AUROC↑ | FPR95↓ | ACC↑ | AURC↓ | AUROC↑ | FPR95↓ | ACC↑ |
| DOCTOR | 2.31 | 94.88 | 27.50 | 96.75 | 18.84 | 89.37 | 59.81 | 89.42 | 481.80 | 72.61 | 89.46 | 31.26 | 85.35 | 88.73 | 55.21 | 70.31 |
| MSP | 2.20 | 95.21 | 26.25 | 96.75 | 18.73 | 89.48 | 58.87 | 89.42 | 485.10 | 72.31 | 90.39 | 31.26 | 84.54 | 88.97 | 56.28 | 70.31 |
| TrustVLM-D | 2.08 | 95.52 | 26.25 | 96.75 | 18.63 | 89.57 | 57.58 | 89.42 | 390.00 | 90.09 | 42.82 | 31.26 | 74.35 | 92.14 | 41.23 | 70.31 |
| TrustVLM*-D | 1.76 | 95.65 | 21.13 | 97.12 | 22.18 | 87.92 | 54.05 | 89.42 | 72.64 | 75.64 | 73.41 | 82.91 | 54.41 | 87.91 | 64.74 | 78.03 |
| | Food101 | | | | SUN397 | | | | Average | | | |
| | AURC↓ | AUROC↑ | FPR95↓ | ACC↑ | AURC↓ | AUROC↑ | FPR95↓ | ACC↑ | AURC↓ | AUROC↑ | FPR95↓ | ACC↑ |
| DOCTOR | 40.90 | 84.68 | 56.72 | 87.54 | 132.64 | 80.73 | 74.81 | 67.62 | 149.27 | 85.69 | 58.63 | 70.69 |
| MSP | 40.70 | 84.84 | 54.70 | 87.54 | 131.29 | 81.15 | 73.63 | 67.62 | 149.13 | 85.86 | 57.31 | 70.69 |
| TrustVLM-D | 22.97 | 89.84 | 54.70 | 87.54 | 117.67 | 85.10 | 59.27 | 67.62 | 129.84 | **90.35** | 44.49 | 70.69 |
| TrustVLM*-D | 23.20 | 89.56 | 56.14 | 87.80 | 97.28 | 83.11 | 73.78 | 72.63 | **87.35** | 87.47 | 53.97 | **79.99** |

Table 11: Misclassification detection performance on fine-grained classification datasets with SigLIP ViT-B/16.

**MisD performance on ImageNet and its variants under distribution shifts.** To evaluate the robustness of our proposed method to distribution shifts, the visual prototypes were computed using only $N$ samples per class from the ImageNet training split. These prototypes were then applied directly to the ImageNet variants, thereby obviating the need to compute variant-specific prototypes. As illustrated in Tab. 10, our method demonstrates the overall leading performance of all metrics evaluated in this challenging scenario, achieving average improvements of 1.54% in AURC, 1.07% in AUROC, and 4.29% in FPR95 relative to the baseline.

**Different architectures and VLMs.** To demonstrate the versatility of the proposed framework, we evaluated its performance with different architectures and VLMs. Specifically, we replaced the default CLIP ViT-B/16 backbone with CLIP ResNet-50 and SigLIP ViT-B/16, reporting average performance metrics across fine-grained datasets and ImageNet and its variants from Tab. 11 to Tab. 14. Consistent with the observations with CLIP ViT-B/16, our method demonstrates robust compatibility across these varied architectures and VLMs, significantly surpassing the baseline methods in both configurations.

## F FURTHER ANALYSIS

**Clarification on image-to-text and image-to-image similarities.** The image-to-text and image-to-image similarities are complementary to each other. Each of them captures different aspects of the data, and their combination leads to superior performance. For example, visually similar objects like "lemon" and "tennis ball" (both round and yellow) may be hard to distinguish in the image embedding space. Yet, image-to-text similarity, which leverages semantic cues (e.g., "a sour fruit" vs "a sports object"), can help disambiguate them more effectively. Conversely, some subtle visual variations (e.g., among different flower species) may be better captured by image-to-image similarity.

To analyze this, we conducted the following experiment: for each image, we randomly sample one positive (same label) and one negative (different label) image, and compute both image-to-image

| | ImageNet-A | | | | ImageNet-V2 | | | | ImageNet-R | | | |
|---|---|---|---|---|---|---|---|---|---|---|---|---|
| | AURC↓ | AUROC↑ | FPR95↓ | ACC↑ | AURC↓ | AUROC↑ | FPR95↓ | ACC↑ | AURC↓ | AUROC↑ | FPR95↓ | ACC↑ |
| DOCTOR | 299.17 | 79.28 | 72.72 | 47.62 | 124.64 | 83.03 | 67.64 | 67.55 | 21.40 | 91.53 | 45.00 | 87.82 |
| MSP | 299.17 | 79.25 | 72.58 | 47.62 | 124.12 | 83.21 | 67.08 | 67.55 | 20.92 | 91.90 | 43.06 | 87.82 |
| TrustVLM-D | 278.77 | 82.13 | 65.09 | 47.62 | 123.39 | 83.76 | 65.28 | 67.55 | 19.55 | 92.07 | 42.05 | 87.82 |
| TrustVLM*-D | 266.12 | 77.68 | 78.19 | 51.82 | 122.70 | 83.05 | 67.17 | 68.17 | 19.49 | 91.85 | 43.33 | 87.98 |
| | ImageNet-Sketch | | | | ImageNet | | | | Average | | | |
| | AURC↓ | AUROC↑ | FPR95↓ | ACC↑ | AURC↓ | AUROC↑ | FPR95↓ | ACC↑ | AURC↓ | AUROC↑ | FPR95↓ | ACC↑ |
| DOCTOR | 136.52 | 84.98 | 62.35 | 64.29 | 86.80 | 83.40 | 67.59 | 74.74 | 133.71 | 84.44 | 63.06 | 68.40 |
| MSP | 135.32 | 85.31 | 61.24 | 64.29 | 85.84 | 83.78 | 66.03 | 74.74 | 133.07 | 84.69 | 62.00 | 68.40 |
| TrustVLM-D | 133.55 | 85.84 | 59.61 | 64.29 | 84.46 | 84.36 | 63.73 | 74.74 | 127.94 | **85.63** | **59.15** | 68.40 |
| TrustVLM*-D | 132.62 | 84.76 | 64.17 | 65.14 | 83.96 | 83.40 | 67.89 | 75.45 | **124.98** | 84.15 | 64.15 | **69.71** |

Table 12: Misclassification detection performance on ImageNet and its variants with SigLIP ViT-B/16.

| | Flowers102 | | | | DTD | | | | Aircraft | | | | Pets | | | |
|---|---|---|---|---|---|---|---|---|---|---|---|---|---|---|---|---|
| | AURC↓ | AUROC↑ | FPR95↓ | ACC↑ | AURC↓ | AUROC↑ | FPR95↓ | ACC↑ | AURC↓ | AUROC↑ | FPR95↓ | ACC↑ | AURC↓ | AUROC↑ | FPR95↓ | ACC↑ |
| DOCTOR | 154.49 | 83.85 | 67.94 | 61.75 | 374.42 | 75.61 | 83.55 | 40.37 | 695.10 | 74.86 | 79.37 | 15.54 | 36.51 | 87.78 | 62.33 | 83.65 |
| MSP | 154.02 | 83.99 | 69.57 | 61.75 | 374.63 | 75.58 | 83.85 | 40.37 | 693.70 | 75.15 | 80.97 | 15.54 | 35.63 | 88.30 | 59.83 | 83.65 |
| TrustVLM-D | 106.43 | 94.74 | 27.07 | 61.75 | 309.72 | 87.72 | 47.87 | 40.37 | 675.03 | 79.38 | 79.37 | 15.54 | 35.02 | 88.62 | 58.00 | 83.65 |
| TrustVLM*-D | 0.23 | 98.10 | 4.55 | 99.11 | 130.86 | 77.64 | 75.46 | 71.34 | 622.50 | 75.35 | 79.68 | 20.25 | 34.37 | 88.95 | 57.67 | 83.65 |
| | Caltech101 | | | | Cars | | | | EuroSAT | | | | UCF101 | | | |
| | AURC↓ | AUROC↑ | FPR95↓ | ACC↑ | AURC↓ | AUROC↑ | FPR95↓ | ACC↑ | AURC↓ | AUROC↑ | FPR95↓ | ACC↑ | AURC↓ | AUROC↑ | FPR95↓ | ACC↑ |
| DOCTOR | 30.50 | 87.30 | 69.54 | 85.88 | 212.51 | 80.23 | 74.73 | 55.74 | 618.09 | 70.73 | 83.87 | 23.70 | 169.84 | 84.01 | 72.38 | 58.82 |
| MSP | 29.55 | 87.95 | 66.38 | 85.88 | 209.07 | 80.93 | 72.99 | 55.74 | 633.47 | 67.67 | 87.27 | 23.70 | 166.92 | 84.77 | 68.98 | 58.82 |
| TrustVLM-D | 22.17 | 93.16 | 36.49 | 85.88 | 206.48 | 81.70 | 70.75 | 55.74 | 514.75 | 87.02 | 54.70 | 23.70 | 144.99 | 89.67 | 49.71 | 58.82 |
| TrustVLM*-D | 6.06 | 89.61 | 40.91 | 96.43 | 205.62 | 81.41 | 70.92 | 56.08 | 101.58 | 70.14 | 74.00 | 82.43 | 73.84 | 85.69 | 62.81 | 75.34 |
| | Food101 | | | | SUN397 | | | | Average | | | | | | | |
| | AURC↓ | AUROC↑ | FPR95↓ | ACC↑ | AURC↓ | AUROC↑ | FPR95↓ | ACC↑ | AURC↓ | AUROC↑ | FPR95↓ | ACC↑ | | | | |
| DOCTOR | 94.96 | 83.59 | 65.77 | 73.95 | 209.19 | 77.49 | 78.84 | 58.81 | 259.56 | 80.55 | 73.83 | 55.82 | | | | |
| MSP | 94.32 | 83.83 | 64.86 | 73.95 | 206.31 | 78.16 | 76.41 | 58.81 | 259.76 | 80.63 | 73.11 | 55.82 | | | | |
| TrustVLM-D | 76.19 | 87.05 | 58.46 | 73.95 | 178.75 | 84.08 | 57.38 | 58.81 | 226.95 | **87.31** | **53.98** | 55.82 | | | | |
| TrustVLM*-D | 70.64 | 87.03 | 60.27 | 75.23 | 112.43 | 81.19 | 73.47 | 71.45 | **135.81** | 83.51 | 59.97 | **73.13** | | | | |

Table 13: Misclassification detection performance on fine-grained classification datasets with CLIP ResNet-50.

| | ImageNet-A | | | | ImageNet-V2 | | | | ImageNet-R | | | |
|---|---|---|---|---|---|---|---|---|---|---|---|---|
| | AURC↓ | AUROC↑ | FPR95↓ | ACC↑ | AURC↓ | AUROC↑ | FPR95↓ | ACC↑ | AURC↓ | AUROC↑ | FPR95↓ | ACC↑ |
| DOCTOR | 672.36 | 66.71 | 86.78 | 22.79 | 256.91 | 79.22 | 75.41 | 51.50 | 188.11 | 84.28 | 68.02 | 56.36 |
| MSP | 673.06 | 66.68 | 86.39 | 22.79 | 255.65 | 79.53 | 74.83 | 51.50 | 187.18 | 84.51 | 66.21 | 56.36 |
| TrustVLM-D | 629.41 | 72.99 | 75.33 | 22.79 | 242.91 | 82.75 | 62.07 | 51.50 | 172.53 | 87.58 | 56.44 | 56.36 |
| TrustVLM*-D | 583.73 | 67.67 | 85.08 | 27.99 | 231.81 | 77.67 | 75.42 | 56.13 | 166.54 | 83.35 | 71.47 | 60.34 |
| | ImageNet-Sketch | | | | ImageNet | | | | Average | | | |
| | AURC↓ | AUROC↑ | FPR95↓ | ACC↑ | AURC↓ | AUROC↑ | FPR95↓ | ACC↑ | AURC↓ | AUROC↑ | FPR95↓ | ACC↑ |
| DOCTOR | 446.07 | 78.27 | 74.71 | 32.99 | 204.36 | 79.44 | 73.82 | 58.18 | 353.56 | 77.58 | 75.75 | 44.36 |
| MSP | 444.12 | 78.62 | 73.94 | 32.99 | 202.30 | 79.96 | 72.03 | 58.18 | 352.46 | 77.86 | 74.68 | 44.36 |
| TrustVLM-D | 409.65 | 85.09 | 52.28 | 32.99 | 183.69 | 84.24 | 58.35 | 58.18 | 327.64 | **82.53** | **60.89** | 44.36 |
| TrustVLM*-D | 338.03 | 71.57 | 80.89 | 48.05 | 168.30 | 77.91 | 74.34 | 65.00 | **297.68** | 75.63 | 77.44 | **51.50** |

Table 14: Misclassification detection performance on ImageNet and its variants with CLIP ResNet-50.

and image-to-text similarity differences, as shown in Tab. 15. We find that the relative discriminative power varies by dataset. For example, on Flowers102, image-to-image similarity yields larger differences in $97.68\%$ of cases. On Cars, this number drops to $42.99\%$, indicating that image-to-text similarity can be more informative on certain datasets.

**Influences of weights on the confidence-scoring function.** Based on our analysis above, assigning a higher weight to the image-to-image (i-i) similarity term can be beneficial when it provides stronger discriminative signals—for example, on datasets like Flowers102, where visual features are more distinctive. As shown in Tab. 16 and Tab. 17, increasing the i-i weight improves performance on Flowers102 but degrades performance on Cars, where fine-grained classes are better distinguished by image-to-text (i-t) similarity. This observation is consistent with our earlier findings that the relative effectiveness of i-i vs. i-t varies across datasets.

|  | MSP | TrustVLM-D | ratio where $S_{i-i} > S_{i-t}$ |
|---|---|---|---|
| Flower102 | 85.91 | 95.05 | 0.97 |
| DTD | 79.81 | 88.55 | 0.83 |
| Aircraft | 72.62 | 75.62 | 0.63 |
| Pets | 89.94 | 90.05 | 0.70 |
| Caltech101 | 86.99 | 90.51 | 0.98 |
| Cars | 81.95 | 82.05 | 0.42 |
| EuroSAT | 76.39 | 85.48 | 0.83 |
| UCF101 | 85.98 | 90.21 | 0.91 |
| Food101 | 84.51 | 88.52 | 0.76 |
| SUN397 | 77.90 | 83.80 | 0.95 |

Table 15: For each image, we randomly sampled one positive example (same class) and one negative example (different class), and computed both $S_{i-i}$ and $S_{i-t}$. We then measured the proportion of cases where the $S_{i-i}$ exceeds $S_{i-t}$. This ratio directly quantifies the relative discriminative strength of the visual features compared to the text-aligned semantic features. The AUROC is reported.

| Weight | AURC↓ | AUROC↑ | FPR95↓ |
|---|---|---|---|
| 0.2 | 100.53 | 88.68 | 54.41 |
| 1.0 | 77.30 | 95.05 | 30.06 |
| 2.0 | 67.19 | 97.96 | 12.55 |

Table 16: Influence of different weights on the confidence-scoring function (TrustVLM-D on Flowers102 with N=16).

| Weight | AURC↓ | AUROC↑ | FPR95↓ |
|---|---|---|---|
| 0.2 | 135.21 | 82.26 | 71.92 |
| 1.0 | 137.54 | 82.05 | 70.62 |
| 2.0 | 148.86 | 79.82 | 72.76 |

Table 17: Influence of different weights on the confidence-scoring function (TrustVLM-D on Cars with N=16).

| Prompt | AURC↓ | AUROC↑ | FPR95↓ |
|---|---|---|---|
| A photo of a [class] | 77.30 | 95.05 | 30.06 |
| A figure of a [class] | 73.96 | 95.30 | 26.08 |
| An image of a [class] | 70.32 | 95.55 | 24.90 |

Table 18: Influence of different text prompts (TrustVLM-D on Flowers102 with N=16).

In our experiments, we find that using equal weights (i.e., weight = 1.0 for both terms) yields strong and stable performance across diverse datasets, without the need for dataset-specific tuning. This simple uniform weighting offers a good balance between robustness and generality, though we agree that adaptive weighting based on dataset characteristics could be a promising future direction.

**Influences of different text prompts.** In our experiments, we use the default CLIP-style prompt "A photo of a [class]". To assess the robustness of our method to prompt variations, we replaced it with alternative prompts such as "A figure of a [class]" and "An image of a [class]". As shown in the results from Tab. 18 to Tab. 20, the performance remains stable—and in some cases even improves—with these alternative prompts. This suggests that our framework is robust to reasonable prompt variations and does not rely heavily on a specific prompt template. We believe this robustness stems from the way our method estimates confidence based on both image-to-text and image-to-image similarity, rather than being overly sensitive to minor linguistic changes in the input text prompts.

**Robustness to spurious correlation.** To investigate the robustness of our framework to spurious correlations, we experimented on the Waterbirds dataset (Sagawa et al., 2019). Using N-shot

| Prompt | AURC↓ | AUROC↑ | FPR95↓ |
|---|---|---|---|
| A photo of a [class] | 562.02 | 75.62 | 83.21 |
| A figure of a [class] | 576.58 | 77.08 | 77.34 |
| An image of a [class] | 566.30 | 77.41 | 80.70 |

Table 19: Influence of different text prompts (TrustVLM-D on Aircraft with N=16).

| Prompt | AURC↓ | AUROC↑ | FPR95↓ |
|---|---|---|---|
| A photo of a [class] | 11.11 | 90.51 | 47.27 |
| A figure of a [class] | 9.42 | 92.71 | 42.39 |
| An image of a [class] | 7.39 | 93.36 | 41.10 |

Table 20: Influence of different text prompts (TrustVLM-D on Caltech101 with N=16).

| | AURC↓ | AUROC↑ | FPR95↓ |
|---|---|---|---|
| MSP | 76.73 | 76.65 | 84.35 |
| TrustVLM-D | 70.18 | 79.80 | 76.57 |

Table 21: Robustness to spurious correlation on Waterbirds dataset.

samples from the (spurious-biased) training split to build our visual prototypes, TrustVLM-D still outperforms MSP, although the gap narrows. We attribute this to a domain shift introduced by the spurious correlation between training prototypes and test images. We then select N-shot samples from the testing data to calculate visual prototypes and evaluate on the remaining test data. As shown in Tab. 21, without a domain shift between visual prototypes and testing data, our TrustVLM-D significantly outperforms the baseline. This confirms that, even under pronounced spurious correlations, our method's reliance on complementary embedding spaces can robustly distinguish correct from incorrect predictions, so long as prototype and evaluation domains align.

**Comparison of different auxiliary vision encoders.** DINOv2's strong performance can be attributed to a combination of training methodology and training data. DINOv2 uses a self-distillation without labels framework with ViT backbones, encouraging the model to learn semantically rich and spatially coherent features. This typically results in embeddings that generalize well across diverse downstream tasks. DINOv2 is trained on a large and diverse dataset without human-annotated labels, which helps it capture generic visual patterns useful across many domains. We observe that on fine-grained datasets like Pets and Cars, MoCo v2 and CLIP-I sometimes outperform DINOv2. This suggests that CLIP-I, trained with image-text alignment, may emphasize semantic-level features that are more aligned with class labels defined by textual concepts (e.g., specific car models or pet breeds). MoCo v2, trained with contrastive learning on ImageNet, might preserve low-level visual cues better than DINOv2, which can help in datasets where subtle details (e.g., fur texture, head shape) are critical for class discrimination. These differences imply that each vision encoder has different feature biases, depending on what features they emphasize in their embedding space. DINOv2 is strong on semantic abstraction and global structure. CLIP-I is strong on semantic alignment with language. MoCo v2 is strong on local patterns and fine-grained visual features. This suggests that the choice of the auxiliary vision encoder can impact performance in a dataset-dependent way, and a promising direction is to adaptively choose or fuse multiple encoders depending on the domain characteristics.

**Incorporating more than one auxiliary vision encoder.** We incorporate both CLIP-I and DINOv2 as auxiliary vision encoders for experiments. Specifically, we construct two separate sets of visual prototypes—one from each encoder—and compute two image-to-image similarity scores, which are then summed to obtain the final confidence score. Our experiments show that combining multiple vision encoders improves misclassification detection performance in most cases. For example, on Flowers102, the combined model achieves 74.21 on AURC, 95.94 on AUROC, and 23.60 on FPR95, all better than using either CLIP-I or DINOv2 alone. Similarly, on UCF101, it achieves 104.26 on AURC, 91.04 on AUROC, and 45.14 on FPR95, again outperforming single-encoder baselines. These results suggest that different vision encoders capture complementary visual features, and combining them leads to more robust and reliable confidence estimation. This points to an exciting direction

for future work — further exploring encoder fusion strategies for enhanced misclassification and uncertainty detection.

**Comparison with large vision-language models (LVLMs) for confidence estimation.** We conducted experiments using Qwen2.5-VL-3B-Instruct (Bai et al., 2025) to evaluate whether it can assess the validity of predictions made by a custom VLM. For each test image, we provided the following prompt:

"You are given an image and a predicted label from a vision-language model.

Predicted label: "{predicted_label}"

Please answer the following: On a scale from 0 to 100, how confident are you that this label is correct? (0 = not confident at all, 100 = completely confident)".

We find that Qwen2.5-VL produces reasonable confidence estimates. On the AUROC metric, it achieves 73.05 on Flowers102, 71.20 on Cars, 72.62 on UCF101, and 82.77 on Caltech101. These results are promising and demonstrate that LVLMs can serve as an alternative way to estimate prediction confidence. However, our proposed framework still outperforms Qwen2.5-VL across all datasets, highlighting the advantage of our tailored design for reliable confidence estimation. Moreover, our method is more lightweight and data-efficient, requiring neither large-scale instruction tuning nor expensive inference. This experiment validates the potential of LVLMs in this space and opens up exciting directions for future research, such as integrating LVLM-based reasoning into confidence estimation pipelines.

**Comparison with Monte Carlo dropout and data augmentation for confidence estimation.** We first explored Monte Carlo (MC) Dropout (Gal & Ghahramani, 2016) for estimating epistemic uncertainty. However, the CLIP ViT-B/16 model does not include dropout layers in its transformer blocks by default. To enable MC Dropout, we inserted dropout into the MLP and attention layers and performed 64 stochastic forward passes. We then computed uncertainty using the variance of the predicted class probabilities. MC Dropout showed limited effectiveness in detecting misclassifications in our experiments. For example, on the Flowers102 dataset, it achieved an AURC of 408.38, AUROC of 47.65, and FPR95 of 95.72—significantly worse than our proposed method and other baselines. On UCF101, the results were similarly poor: AURC 510.05, AUROC 42.54, and FPR95 97.54. We hypothesize that this degradation stems from two issues: (1) CLIP's pretrained transformer architecture is not optimized for stochastic perturbations, and inserting dropout disrupts its learned representations; (2) MC Dropout's predictive variance is not well-calibrated in high-dimensional vision-language settings, making it less effective for uncertainty estimation in multimodal models like CLIP.

We then experimented with using data augmentation to estimate aleatoric uncertainty. Specifically, we generated 64 augmented views of each input image using standard techniques such as random rotation, translation, resized cropping, and horizontal flipping. We then computed the variance of the predicted class probabilities across these augmented views as a measure of uncertainty. However, this approach also did not perform well in detecting misclassifications in our experiments. For example, on the Flowers102 dataset, it achieved an AURC of 314.75, AUROC of 52.89, and FPR95 of 96.06—significantly worse than our proposed method and other baselines. On UCF101, the results were similarly poor, with AURC of 415.78, AUROC of 43.36, and FPR95 of 97.47. We believe the limited performance stems from two main challenges: (1) Data augmentation primarily captures input noise (aleatoric uncertainty), which does not sufficiently explain the model's confidence on out-of-distribution or hard in-distribution examples; (2) CLIP's zero-shot predictions tend to be highly stable across augmented views, which can lead to low measured variability even on incorrect predictions, thus reducing the effectiveness of augmentation-based uncertainty estimation.

## G  DISCUSSION ON THE POTENTIAL REASONS BEHIND THE ROBUSTNESS OF TRUSTVLM

In multimodal learning, if one modality is significantly less reliable or noisy, a naive ensemble of confidence scores could allow the noisy signal to degrade the high-quality signal. In our experiments, we showed that TrustVLM remains robust even under severely mismatched or imbalanced modality pairs. For severely mismatched modality pairs, we simulate scenarios where the visual prototypes

are intentionally degraded while the image-text is well aligned. This creates a controlled mismatch between image embeddings and visual prototypes. We evaluate TrustVLM under two visual prototype degradation scenerios: synthetic-real shift (Tab. 9) and distribution shift (Tab. 10). These results demonstrate that TrustVLM's ensemble score remains stable even under severely mismatched modality conditions. Results in Tab. 15 demonstrate that TrustVLM consistently outperforms baselines under imbalanced modality pairs where one modality is inherently more discriminative than the other. Below, we discuss the potential reasons behind this behavior.

$S_{i-i}$ **acts as a verification signal.** $S_{i-i}$ is explicitly conditioned on the VLM's prediction $\hat{y}$. If $\hat{y}$ is wrong, then $E_x$ is compared to the wrong class prototype $P_{\hat{y}}$, which is typically far in the embedding space and $S_{i-i}$ tends to be low. If $\hat{y}$ is correct, then $E_x$ is compared to the right prototype $P_{\hat{y}}$, which is close and $S_{i-i}$ tends to be high. Thus, $S_{i-i}$ behaves as a consistency check on $S_{i-t}$. When both $S_{i-i}$ and $S_{i-t}$ are high (Agreement), $\kappa$ increases and the prediction appears trustworthy. When one is high and the other is low (Disagreement), $\kappa$ decreases and the prediction is flagged as risky. This is quite different from ensembling arbitrary scores from unrelated modalities (like tactile–thermal), where there is no such *verify the predicted class* structure.

**MisD metrics depend on ranking, not absolute values.** MisD metrics (AUROC, AURC, FPR@95) depend on relative ranking of $\kappa$ between correct and wrong predictions, not the absolute scale. As long as, on average, $\mathbb{E}[S_{i-i} \mid \text{correct}] \gtrsim \mathbb{E}[S_{i-i} \mid \text{wrong}]$, adding $S_{i-i}$ to $S_{i-t}$ tends to improve or preserve the ranking. Correct predictions receive a positive contribution from $S_{i-i}$, while wrong predictions receive little or even negative contribution. This widens the separation between the score distributions of correct and incorrect predictions, thereby improving MisD performance.

**Averaging over prototypes mitigates severe mismatch.** Prototypes are averages over N-shot embeddings per class. Thus, even if some training images are noisy, or the auxiliary encoder has a domain bias, the class prototype $P_c$ lies near the center of the class cluster, not at single outlier points. This means for a correct test image, similarity to $P_{\hat{y}}$ is typically higher than to other prototypes, even under shift. For a wrong prediction, similarity to the wrong class prototype tends to be much lower. So even in the presence of moderate mismatch, $S_{i-i}$ retains the *correct > wrong* tendency at the class level, which is exactly what the MisD ranking needs.

