# OpenReview forum: "To Trust Or Not To Trust Your Vision-Language Model's Prediction"
_ICLR.cc/2026/Conference — Submitted to ICLR 2026_

### Official Review · Reviewer_aUQM · 2025-10-28

**Soundness:** 2
**Presentation:** 3
**Contribution:** 1
**Rating:** 2
**Confidence:** 4

**Summary:**

This paper introduces TrustVLM, a framework designed to address confidence estimation in vision-language model (VLM) predictions. Motivated by the modality gap inherent in VLMs, TrustVLM constructs ensembles of VLMs and image-only classifiers to enhance the detection of misclassified samples.

**Strengths:**

The presentation of this paper is clear and easy to follow.

**Weaknesses:**

The methodology proposed in this paper employs N-sample training data together with external image encoders to construct a Nearest Class Mean classifier, which is then combined with the original CLIP classifier. However, the improvement appears to stem primarily from the use of additional labeled data and model ensembling. I am concerned that this may not constitute a genuinely novel contribution. Moreover, since the competitive baselines operate in a zero-shot setting, the comparison could be considered unfair.

**Questions:**

How does TrustVLM perform with fewer training samples, such as in 1-shot or 2-shot settings? Can TrustVLM function with little or even no training data?

---

> ### Author Response · Authors · 2025-11-19
>
> Thanks for your insightful reviews! We provide the responses to your questions as follows:
>
> >**Q1**: How does TrustVLM perform with fewer training samples
>
> **A1**: Thank you for this important question. We investigated this precise point, and our analysis confirms that **TrustVLM is highly effective in low-data regimes, including 1-shot settings**. As detailed in Figure 8, our framework's performance using just a single labeled sample per class (N=1) already achieves results superior to the zero-shot MSP baseline. The performance continues to improve steadily with 2-shot and 4-shot samples, with the gains starting to saturate after N=4. As shown in Tables 7 and 8, the performance remains consistently strong and stable even when using different, randomly selected single samples per class. These findings demonstrate that our method is highly data-efficient. The practical cost is not a large training dataset but merely a single representative example per class, a standard and minimal assumption for few-shot learning. This highlights our **core contribution: a novel scoring function that effectively fuses semantic and visual information to extract a significant boost in reliability from minimal data**.
> ___
> >**Q2**: Can TrustVLM function with little or even no training data
>
> **A2**: Thank you for the question. Yes, our framework is designed to be highly data-efficient and **can function in a "zero-real-data" setting**.
> We explicitly evaluated this scenario in Table 9. In this experiment, we **used a generative model (Stable Diffusion 3) to synthesize visual prototypes** for the UCF101 dataset, completely avoiding the need for any real labeled data.
> The results show that this synthetic-data variant, TrustVLM-D (SD3), still outperforms the strong MSP baseline. This demonstrates a practical path for deploying TrustVLM even in data-scarce or privacy-sensitive environments where real training samples are not available.
>
> ___
> >**Q3**: novelty and unfair comparison
>
> **A3**: We thank the reviewer for this insightful comment. We would like to clarify a potential misunderstanding regarding the paper's core contribution.
>
> The primary novelty of TrustVLM is not to propose a new classification method, but rather **a new confidence scoring function specifically designed for misclassification detection** (MisD). The reviewer is correct that we use N-shot samples to build class prototypes, but their purpose is not simply to create a Nearest Class Mean classifier for ensembling. Instead, they are essential components of our novel scoring function.
>
> Our key insight is that the standard image-to-text (i-t) similarity score (like MSP) is an insufficient and often unreliable metric for a VLM's confidence. Our novelty lies in proposing a new function, $\kappa(x) = S_{i-t} + S_{i-i}$, which fuses the VLM's standard semantic score ($S_{i-t}$) with a complementary visual verification score ($S_{i-i}$) derived from the image embedding space. This fusion is the central contribution, and its effectiveness is validated by our significant performance gains—up to 51.87% in AURC, 9.14% in AUROC, and 32.42% in FPR95 over strong baselines.
>
> Regarding the concern about fairness and labeled data, we would like to highlight two key points:
>
> **1. Our framework is highly data-efficient.** The use of N-shot samples is intentionally minimal. As demonstrated in Figure 8, TrustVLM significantly outperforms strong zero-shot baselines even with only a single labeled sample (N=1). We even demonstrate in Table 9 that TrustVLM outperforms the strong MSP baseline using only synthetically generated data. This establishes a practical and data-efficient benchmark, not an unfair one.
>
> **2. Competitive baselines are not all zero-shot.** We disagree that the comparison is unfair, as recent state-of-the-art methods also leverage external resources. For instance, in Table 3, we compare against ORCA, which **requires prompting large language models (e.g., GPT-3.5) to construct a concept collection**. Despite ORCA's reliance on this extensive external knowledge, TrustVLM outperforms it across all reported datasets and backbones.
>
> *In summary, our contribution is a novel and highly effective MisD framework that leverages minimal data (as little as N=1 or even synthetic data) to fuse complementary semantic and visual information. This approach is rigorously validated across 17 datasets, 4 architectures, and 2 VLMs, demonstrating its effectiveness and broad applicability.*
>
> We believe these clarifications strengthen the case for TrustVLM's contributions and hope they address your concerns.
>
> ___
> We sincerely thank the reviewer again for the constructive feedback. If there are any additional questions or concerns, we would be more than happy to further address them.

---

> ### Comment · Reviewer_aUQM · 2025-11-26
>
> Thanks for your response.
>
> > Our framework is highly data-efficient.
>
> After reviewing the results provided by the authors regarding the number of labeled samples, as well as their experiments on synthesizing images in a no-data scenario, my concerns about data efficiency have been addressed.
>
> > Competitive baselines are not all zero-shot.
>
> In my view, leveraging external resources — including prompting LLMs or synthesizing images — can still be considered consistent with zero-shot methodology, as long as no target-domain data (labeled or unlabeled) is required.
>
> > Our novelty lies in proposing a new function, which fuses the VLM's standard semantic score with a complementary visual verification score derived from the image embedding space.
>
> I am afraid I cannot fully accept this as a novelty claim.
>
> 1. **Combining visual and semantic scores has been extensively explored.**
>    In few-shot CLIP adaptation [1,2], prior work has already investigated integrating visual similarity with image–text similarity using few-shot samples.
>
> 2. **Combining CLIP with external vision encoders is also not new.**
>    Several existing works [3,4] study the integration of CLIP with other visual models for vision–language tasks, demonstrating that visual verification or fusion-based strategies have prior foundations.
>
> This work appears to be a direct combination of the two aforementioned approaches, and it is not immediately clear if it offers significant insights beyond a straightforward ensemble. As a result, I find the novelty somewhat limited.
>
> **References:**
> [1] Zhang, Renrui, et al. *"Tip-Adapter: Training-free Adaptation of CLIP for Few-shot Classification."* ECCV 2022.
> [2] Wang, Zhengbo, et al. *"A Hard-to-Beat Baseline for Training-free CLIP-based Adaptation."* ICLR 2024.
> [3] Jiang, Dongsheng, et al. *"From CLIP to DINO: Visual Encoders Shout in Multi-Modal Large Language Models."* arXiv:2310.08825 (2023).
> [4] Imam, Mohamed Fazli, et al. *"CLIP Meets DINO for Tuning Zero-Shot Classifier Using Unlabeled Image Collections."* arXiv:2411.19346 (2024).

---

> > ### Author Response · Authors · 2025-11-27
> >
> > Thank you for your thoughtful and constructive engagement with our work!
> >
> > >**Q1**: Fairness of Comparison and Zero-Shot Definition
> >
> > **A1**: We thank the reviewer for this constructive clarification. We fully appreciate your view that utilizing external resources (like synthesized images or LLMs) constitutes a "zero-shot" setting provided no target-domain data is accessed.
> > We agree with this definition. Based on this criterion, **TrustVLM is in fact capable of operating as a state-of-the-art zero-shot method**, and our comparisons are fair and robust. We address this through three key evidence points below:
> >
> > **1. TrustVLM acts as a Zero-Shot Method via Image Synthesis (Table 9).** The reviewer correctly notes that "synthesizing images" is consistent with zero-shot methodology. As demonstrated in Table 9, we explicitly validated a variant of our framework, TrustVLM-D (SD3), which generates prototypes using Stable Diffusion 3 rather than real data. This setup accesses zero real images (labeled or unlabeled). Even in this strict zero-shot setting, TrustVLM-D (SD3) still outperforms the strong MSP baseline. This confirms that **our framework is effective without requiring any real data**, ensuring a fair comparison against other zero-shot baselines.
> >
> > **2. Proven Robustness without Target-Domain Data (Fig. 3 and Table 10).** The reviewer emphasized that fairness relies on **not using target-domain data**. We extensively validated our method under this exact constraint via our distribution shift experiments. In Figure 3 and Table 10, we construct prototypes exclusively using standard ImageNet training images and evaluate directly on challenging, shifted variants (ImageNet-A, -R, -Sketch, -V2). **TrustVLM consistently outperforms baselines on these target domains without having seen a single sample (labeled or unlabeled) from the target distribution.** This demonstrates that our method does not rely on target-domain statistics to be effective, satisfying the reviewer's criterion for a fair evaluation against zero-shot baselines.
> >
> > **3. Practicality and the Safety Trade-off.** Misclassification Detection (MisD) for VLMs is **a relatively new field with limited existing baselines**; thus, comparing against zero-shot methods (like MSP or ORCA) is necessary to benchmark progress. However, we believe TrustVLM offers a unique value proposition:
> >
> > *Efficiency:* We demonstrate that using just one image per class ($N=1$) (a highly reasonable assumption for safety-critical applications) yields state-of-the-art performance.
> >
> > *Flexibility:* Our framework offers a spectrum of deployment options, from strict zero-shot (using synthetic data or source-domain prototypes) to highly efficient few-shot (using $N=1$ real samples).
> >
> > **Conclusion.** TrustVLM is not just a few-shot method comparing itself to zero-shot baselines. It is a flexible framework that outperforms baselines even when restricted to the reviewer’s definition of zero-shot (via synthetic data or cross-domain transfer).
> > We hope these results regarding the synthetic data and distribution shifts clarify that our comparisons are fair and that the method contributes significantly to VLM safety.

---

> > ### Author Response · Authors · 2025-11-27
> >
> > >**Q2**: Novelty and Differentiation from Adaptation/Ensemble Methods
> >
> > **A2**: We thank the reviewer for the detailed comment and for pointing us to [1-4]. We agree that both (i) combining visual and semantic information and (ii) using CLIP together with other vision encoders are important and active lines of work. However, **our contribution is not a generic few-shot adaptation or ensemble method, but a training-free misclassification detection (MisD) framework tailored to VLMs, which is not addressed in [1–4]**. Below we clarify the conceptual and technical differences.
> >
> > **1. Relation to few-shot CLIP adaptation [1,2]**
> >
> > **Different problem and objective.** Tip-Adapter and “A Hard-to-Beat Baseline” are designed to improve classification accuracy of CLIP in few-shot or training-free adaptation settings. They build visual caches or GDA classifiers on CLIP features and then ensemble these with the zero-shot text classifier to obtain better predictions on downstream tasks. In contrast, **our primary goal is misclassification detection for VLMs**: given any prediction from a (possibly already adapted) VLM, **we estimate whether this prediction should be trusted or rejected**. Our main metrics are AURC, AUROC, and FPR95, not top-1 accuracy, and a large part of the paper is devoted to analyzing why existing confidence/OOD scores fail in this MisD setting and how to design a better confidence-scoring function.
> >
> > **How our use of visual prototypes differs.** Tip-Adapter builds a key-value cache and linearly combines cache responses with CLIP’s logits to form a new classifier. The “Hard-to-Beat Baseline” uses Gaussian Discriminant Analysis over visual features and ensembles the resulting classifier with the zero-shot text classifier. By contrast, in TrustVLM:
> >
> > 1. We first let the VLM produce its usual zero-shot prediction via image–text similarity (image-to-text, $S_{i-t}$).
> >
> > 2. We then verify this single predicted label using image-to-image similarity between the test image and class prototypes from an auxiliary vision encoder (our $S_{i-i}$ term).
> >
> > 3. The combined trust score $\kappa(x) = S_{i-t} + S_{i-i}$ is used only as a reliability signal, not as an alternative classifier.
> >
> > This **“verify-the-prediction” design** is fundamentally different from building a second classifier and averaging its logits. It explicitly leverages the modality gap and the empirical observation that some concepts are better separated in the image embedding space than in the text space (Fig. 1 and Sec. 3.1). The prototypes thus play the role of an independent second opinion for trust estimation.
> >
> > **Non-trivial beyond a simple ensemble.** We ablate the components in Tab. 6: using only image–text similarity (“i-t”, essentially MSP) or only image–image similarity (“i-i”) both perform clearly worse than our specific combination, confirming that the improvement is not from naively ensembling two scores but from a deliberately designed two-step prediction + verification pipeline. Moreover, we show that standard OOD scores adapted from vision literature (MaxLogit, Energy, Entropy, etc.) are consistently weaker than MSP for MisD, and that our score significantly and consistently improves over all of them on 17 datasets and multiple VLMs (Tabs. 1–3). These MisD-specific insights are, to our knowledge, not present in [1,2].
> >
> > Finally, our framework is orthogonal to [1,2]: in principle, TrustVLM could be applied on top of any CLIP adaptation method (including Tip-Adapter or the GDA baseline) as an external trust module for their predictions.
> >
> > **2. Relation to combining CLIP with other visual encoders [3,4]**
> >
> > We fully agree that coupling CLIP with other vision encoders, especially DINO, is an active topic. “From CLIP to DINO” studies which visual encoders work best inside multi-modal LLMs and proposes COMM, a feature-merging strategy to enhance visual capabilities for tasks such as captioning, VQA, and grounding. “CLIP meets DINO” proposes NoLA, a label-free method that uses DINO features plus LLM-generated textual descriptions to tune CLIP’s zero-shot classifier using unlabeled images, again with the main goal of improving classification performance.
> >
> > Our use of external vision encoders is substantially different:
> >
> > **Role of the auxiliary encoder.** We do not fuse CLIP and DINO (or MoCo) features to build a better classifier, nor do we tune CLIP with DINO supervision. The auxiliary encoder is only used to compute class prototypes and image-to-image similarity, which serve as a trust signal to detect overconfident mistakes. CLIP itself (or SigLIP) remains frozen.
> >
> > **MisD-focused analysis.** The contribution of the auxiliary encoder is analyzed specifically through MisD metrics (AURC, AUROC, FPR95) and the separation of confidence distributions between correct vs. wrong predictions (Fig. 4–5), rather than through improvements in downstream accuracy alone.

---

> > ### Author Response · Authors · 2025-11-27
> >
> > >**Q3**: Insights behind TrustVLM
> >
> > **A3**: Below, we clarify the specific insights that distinguish TrustVLM from a straightforward ensemble:
> >
> > **1. Exploiting the Modality Gap.** Standard ensembles typically combine models to reduce variance. In contrast, TrustVLM is motivated by the insight that VLMs suffer from a "modality gap", where image and text embeddings reside in distinct regions of the shared representation space.
> >
> > **Insight:** Image and text embeddings live in different regions of the joint space, and some concepts are actually more separable in the image embedding space than in the text space.
> >
> > **Evidence:** As shown in Figure 1, the separation margin between concepts like "dog" and "seaplane" is significantly higher in image-to-image similarity ($0.29$) compared to the standard image-to-text similarity ($0.13$). TrustVLM is designed to retrieve this discriminative visual clue that is explicitly lost during the text-alignment process.
> >
> > **2. A Mechanism for Verification, Not Just Prediction.** While ensembles usually aim to improve top-1 accuracy, TrustVLM specifically targets Misclassification Detection (MisD) by functioning as a verification mechanism against overconfidence.
> >
> > **Insight:** VLMs are prone to "confident errors", where an image is visually aligned with the wrong text caption (high $S_{i-t}$). TrustVLM uses the visual prototype distance ($S_{i-i}$) to "verify" the semantic prediction.
> >
> > **Evidence:** Our results show that when a prediction is wrong, the input image’s embedding ($E_x$) typically has low similarity to the visual prototype of the predicted class, effectively flagging the error. As illustrated in our qualitative analysis, this mechanism separates the score distributions of correct and incorrect predictions much more effectively than standard baselines like MSP.
> >
> > **3. Asymmetry in Discriminative Power.** Our extensive analysis reveals that the visual and text spaces are complementary, not redundant, which is why the "ensemble" effect is so significant here.
> >
> > **Insight:** The relative importance of the two signals varies by dataset type.
> >
> > **Evidence:** In Table 15, we show that for datasets with distinct visual features like Flowers102, image-to-image similarity yields larger discriminative differences in 97.68% of cases. However, for datasets like StanfordCars, where semantic labels (e.g., model years) are critical, the text alignment remains more informative. TrustVLM succeeds because it leverages this structural asymmetry, using the auxiliary vision encoder to capture fine-grained patterns that the semantic encoder may miss.
> > ___
> > We sincerely thank the reviewer again for the constructive feedback. If there are any additional questions or concerns, we would be more than happy to further address them.

---

> ### Author Response · Authors · 2025-11-27
>
> **3. Novelty beyond “a straightforward ensemble”**
>
> We acknowledge that our final scoring function $\kappa(x) = S_{i-t} + S_{i-i}$ is simple but effective. However, we believe the novelty lies in:
>
>
> **Problem focus:** first systematic study of training-free misclassification detection for large VLMs using both image–text and image–image information, across 17 datasets, 4 architectures, and 2 VLMs.
>
>
> **Conceptual insight:** identifying and exploiting the modality gap to use image-space prototypes as a verification signal for VLM predictions, rather than as another classifier.
>
> **Empirical findings:** (i) MSP being a surprisingly strong baseline vs. advanced OOD methods for VLM MisD, and (ii) consistent large gains (up to 51.87\% AURC, 9.14\% AUROC, 32.42\% FPR95) over the strongest existing MisD baselines, including concept-based ORCA, without any retraining of the VLM.
>
>
> **Additional benefit:** the same prototypes can optionally be fine-tuned cheaply, yielding competitive or better classification accuracy than state-of-the-art training-free adaptation methods (Tab. 4), while still supporting MisD.
>
> We hope this clarification addresses the novelty concerns and shows that TrustVLM provides new conceptual and practical insights on when VLM predictions can be trusted, which are complementary to prior work on CLIP adaptation and CLIP–DINO fusion.

---

### Official Review · Reviewer_FUYk · 2025-10-31

**Soundness:** 3
**Presentation:** 3
**Contribution:** 3
**Rating:** 6
**Confidence:** 4

**Summary:**

This paper studies VLM error detection to enhance its trustworthiness. The proposed TrustVLM leverages the multimodal similarity of feature representations to decide whether the current prediction is correct. Particularly, it compares the query example with other examples in both image representation similarity and text representation similarity. The final confidence score for error detection is computed by combining both similarities, thus incorporating multimodal information. Through extensive experiments of quantitative analysis and qualitative study, the effectiveness of the TrustVLM has been superior to most of the baseline methods.

**Strengths:**

- This paper is easy to follow, the motivation is very clear, and the intuition is quite straightforward.
- The proposed TrustVLM is training-free and efficient to deploy. It can also be easily adopted by any VLM architectures.
- The experimental performance is quite promising.

**Weaknesses:**

- The major concern is missing the comparison with unimodal detection methods. The proposed method combines multimodal information to detect prediction errors; however, in the ablation study, there is no comparison with image-only or text-only detection. In this way, it would be clearer which branch of modality would contribute more to the overall performance improvement.
- Moreover, the performance of TrustVLM highly relies on the performance of the employed VLMs; if the VLMs cannot provide high-quality representations, the error detection would be limited.
- Another concern is that due to the existence of a modality gap, the cross-modal similarity could be unstable compared to image-to-image similarity. The misaligned cross-modal pairs would also mislead the error detection.
- After detection, the proposed TrustVLM cannot further rectify the error predictions by finding the correct one.

**Questions:**

- Which branch of modality contributes more to the overall performance improvement? It would be helpful to conduct an ablation study to verify the unimodal detection versus multimodal detection. Moreover, what if we directly ask LLMs to detect the prediction error? As done in ``Machine Vision Therapy: Multimodal Large Language Models Can Enhance Visual Robustness via Denoising In-Context Learning, in ICML 2024'', they leverage the prediction of LLMs to find prediction errors of VLMs, and LLMs can further correct the prediction to find the correct one.
- How would the misaligned multimodal pairs mislead the overall detection performance?
- The acquisition of prototypes could be difficult sometimes. Can the proposed method perform without prototypes?

---

> ### Author Response · Authors · 2025-11-19
>
> Thanks for your insightful reviews and great support of our paper! We provide the responses to your questions as follows:
>
> >**Q1**: Comparison with unimodal detection methods
>
> **A1**: Thank you for this valuable suggestion. We would like to clarify that this exact unimodal comparison is provided in Table 6 of the paper. This ablation study directly evaluates the contribution of each component: The 'i-t' row represents the "text-only" (or more accurately, the VLM's standard text-semantic) unimodal baseline, which is equivalent to MSP. The 'i-i' row represents the "image-only" unimodal baseline, using only the confidence score derived from our proposed image-to-image similarity module. The final row shows the performance of our full multimodal method, which fuses both signals. As the table shows, relying only on image-to-text similarity (i-t, the MSP baseline) is suboptimal. Relying only on image-to-image similarity (i-i) is also suboptimal. Our full method, which combines both, consistently achieves the best performance.  This result clearly demonstrates that **the two branches capture complementary information, and their fusion is critical to the performance improvement**.
> ___
> >**Q2**: Reliance on VLM Quality
>
> **A2**: We thank the reviewer for this point and agree that the quality of the VLM's underlying representations will naturally influence the absolute performance of any misclassification detection (MisD) method. A method cannot be expected to find reliable signals in a completely non-informative representation space.
>
> However, the goal of our work is to more reliably detect when a given VLM is failing, using the very representations it provides. The critical comparison, therefore, is against other MisD baselines (like MSP) that operate on the exact same VLM and are subject to the exact same limitation.
>
> Our experiments were designed to demonstrate that TrustVLM is a general and robust framework that enhances reliability across a wide range of VLMs, regardless of their specific architecture. As shown in Table 5 and the detailed results in Tables 11–14, TrustVLM consistently outperforms the baselines across **Multiple VLM families** (e.g., CLIP and SigLIP) and **Diverse backbones** (e.g., ViT-B/16, ViT-B/32, ResNet-50, and ResNet-101).
>
>
> These results confirm that while the absolute performance ceiling is set by the VLM, our method provides a consistent and significant improvement in reliability relative to the baselines for each of those VLMs.
>
> ___
> >**Q3**: Modality Gap and Misaligned Pairs
>
> **A3**: We thank the reviewer for this precise observation. This instability in cross-modal similarity, often caused by the modality gap, is exactly the failure mode TrustVLM is designed to address.
> The reviewer is correct that a misaligned pair, which is precisely what an overconfident, incorrect prediction is, will mislead any detection method that relies solely on that unstable image-to-text ($S_{i-t}$) score (e.g., MSP).
>
> Our method's core contribution is to introduce the image-to-image ($S_{i-i}$) score as a visual self-verification mechanism to counteract this exact instability.
> When a VLM produces a misaligned pair (e.g., a "dog" image is confidently predicted as "seaplane," as in Fig. 1), the $S_{i-t}$ score is spuriously high. However, our $S_{i-i}$ score compares the "dog" image embedding against the "seaplane" visual prototype. This similarity will be very low, as they are visually dissimilar. By fusing these scores ($\kappa(x) = S_{i-t} + S_{i-i}$), the low $S_{i-i}$ score effectively penalizes and corrects the unstable, high $S_{i-t}$ score, thus mitigating the overconfidence.
>
> This mechanism is not misled by the misaligned pair; it actively uses the visual information from the image embedding space to detect and correct for it. Figures 5 and 7 provide concrete illustrations of this process, showing how our $S_{i-i}$ component **reduces the confidence of misaligned predictions and reinforces the confidence of correct ones**.

---

> ### Author Response · Authors · 2025-11-19
>
> >**Q4**: Comparison to LVLMs and Error Rectification
>
> **A4**: We thank the reviewer for this excellent suggestion.We conducted this exact experiment and included the results in Appendix F (Lines 1083-1100). We prompted a powerful LVLM (Qwen2.5-VL-3B-Instruct) to provide a confidence score (from 0 to 100) for the VLM's prediction. The results demonstrated that **our lightweight, training-free method dramatically outperforms the general-purpose LVLM across all testbeds**. For example, on Flowers102, TrustVLM achieves 95.05 AUROC, whereas the LVLM achieves only 73.05 .
> This validates our approach, showing that a specialized, efficient framework is superior for this specific MisD task compared to a general-purpose (and computationally expensive) LVLM.
>
> The reviewer is correct that our paper's scope is focused on misclassification detection, determining when a prediction is untrustworthy, which is a critical and distinct challenge from error correction. In many safety-critical domains, the primary goal is to reliably flag a high-risk prediction for rejection or human review, which is the problem we solve.
> We agree that using LVLMs to correct errors is a valuable and distinct research direction, as explored in the ICML 2024 paper you mentioned. We believe this is an exciting avenue for future work and have now added this to our Future Work discussion in Appendix B (Lines 700-701).
>
> ___
> >**Q5**: Performance Without Prototypes
>
> **A5**: Thank you for this thoughtful question. The visual prototypes are indeed a core component for computing our proposed $S_{i-i}$ score. We address this concern by evaluating two practical scenarios:
>
> **In a strict N=0 setting (no prototypes):** Without any prototypes, the $S_{i-i}$ score cannot be calculated. The framework would then default to using only the $S_{i-t}$ score, which is equivalent to the MSP baseline. As Figure 8 illustrates, the starting point at N=0 represents this baseline performance. Our contribution is the substantial performance jump achieved by adding just one sample (N=1), highlighting our method's high data efficiency.
>
> **In a "zero-real-data" setting (synthetic prototypes):** If collecting real data is infeasible (e.g., due to privacy concerns), we demonstrate a practical alternative. In Table 9, we generated prototypes using Stable Diffusion 3, requiring no real labeled data. Even with these synthetic prototypes, TrustVLM still outperforms the strong MSP baseline. This confirms the framework's value and applicability in data-scarce or privacy-constrained settings.
> ___
> We sincerely thank the reviewer again for the constructive feedback. If there are any additional questions or concerns, we would be more than happy to further address them.

---

> ### Comment · Reviewer_FUYk · 2025-11-26
> **Thank you for the reply**
>
> Dear Authors,
>
> Thank you for addressing each of my concerns. My remaining question is about the claim that cross-modal score can ensemble informations from different modalities, thus won’t be affected by the modality gap.
>
> For example, dog image is well aligned with dog text with a score 0.1, however, dog audio is significantly far away from the dog text, with score 10. Hence, the averaged ensemble score is seriously affected by outlier values, thus the correct score will not be dominant. Instead of intuitive explanation, I think the claim requires further deep-level justification to support this.
>
> Best regards,
> Reviewer.

---

> > ### Author Response · Authors · 2025-11-27
> >
> > Thank you for your thoughtful and constructive engagement with our work!
> >
> > >**Q1**: Justification for Ensembling Cross-Modal Scores
> >
> > **A1**: We thank the reviewer for this thoughtful question.
> > The reviewer’s concern is valid in settings where unnormalized scores like logits or distances can vary wildly, e.g., 0.1 vs. 10. However, within the specific design of TrustVLM, this issue is structurally prevented. In TrustVLM, both $S_{i-t}$ (Image-Text) and $S_{i-i}$ (Image-Image) are **Cosine Similarities** and strictly **bounded to the range $[-1, 1]$**. In the context of CLIP/DINO embeddings for relevant classes, scores typically float between $0.15$ and $0.9$.
> > Because both scores operate on normalized hyperspheres, they share the same scale. It is mathematically impossible for one modality to produce a score of 10 or 100.
> > Therefore, The "outlier dominance" scenario described is structurally impossible in our framework. A noise signal in one modality cannot explode in magnitude to dominate the signal in the other.
> > ___
> > We sincerely thank the reviewer again for the constructive feedback. If there are any additional questions or concerns, we would be more than happy to further address them.

---

> > > ### Comment · Reviewer_FUYk · 2025-11-27
> > > **Further Discussion**
> > >
> > > Thanks for your reply.
> > >
> > > I am still not convinced by this on a philosophical level. It is not about the scale that concerns me, [0.1, 10] can be scaled to [0.01, 1], but the difference is still significant. What I am worried about is the robustness of how this ensemble scores when it faces severely mismatched or imbalanced modality pairs. This problem is very common in multimodal learning, e.g., image-text is very common, but not the other pairs, such as tactile-thermal pairs. There is no promising mechanism in the proposed method to help justify this. Please identify this problem first, and I am looking forward to a more intrinsic justification.
> > >
> > > Best wishes,
> > > Reviewer.

---

> > > > ### Author Response · Authors · 2025-11-28
> > > >
> > > > We thank the reviewer for highlighting this critical point. We fully agree with the reviewer’s perspective that in multimodal learning, if one modality is significantly less reliable or noisy, a naive ensemble could indeed allow the noisy signal to degrade the high-quality signal. In this response, we empirically show that **TrustVLM remains robust even under severely mismatched or imbalanced modality pairs** and discuss the potential reasons behind this behavior.
> > > >
> > > > **1. TrustVLM remains robust under severely mismatched pairs.**
> > > >
> > > > To directly address the concern of severely mismatched modality pairs, we simulate scenarios where the visual prototypes are intentionally degraded while the image-text are well aligned. This creates a controlled mismatch between image embeddings and visual prototypes. We evaluate TrustVLM under two visual prototype degradation scenerios: synthetic-real shift and distribution shift.
> > > >
> > > > **Synthetic-real shift.** In this setting, we use synthetic images from Stable Diffusion to create visual prototypes (TrustVLM-D (SD3)). This introduces a substantial synthetic-real domain gap, making the visual modality ($S_{i-i}$) weaker than the text modality ($S_{i-t}$). Despite this severe mismatch, TrustVLM still outperforms the MSP baseline, as shown in the table below.
> > > >
> > > > |UCF101|AURC$\downarrow$|AUROC$\uparrow$|FPR95$\downarrow$|
> > > > |-|-|-|-|
> > > > |MSP|122.44|85.98|64.89|
> > > > |TrustVLM-D (SD3)|**119.90**|**86.96**|**62.31**|
> > > >
> > > > *Table 1. MisD performance on UCF101 under synthetic-real shifts, where visual prototypes come from synthetic images.*
> > > >
> > > > **Distribution shift.** Here, we compute visual prototypes only on ImageNet images and evaluate directly on shifted variants (ImageNet-A/R/Sketch/V2). This introduces a large distribution gap for ($S_{i-i}$). Even with this mismatch, TrustVLM consistently outperforms baselines:
> > > >
> > > > |FPR95$\downarrow$|ImageNet-A|ImageNet-V2|ImageNet-R|ImageNet-Sketch|ImageNet|
> > > > |-|-|-|-|-|-|
> > > > |DOCTOR|77.76|73.38|60.54|70.94|74.58|
> > > > |MSP|77.76|71.26|59.43|70.29|72.85|
> > > > |TrustVLM-D|**77.20**|**64.82**|**58.09**|**65.59**|**64.47**|
> > > >
> > > > *Table 2. MisD performance under distribution shifts, where prototypes are computed only on ImageNet but applied to variant datasets.*
> > > >
> > > > These results demonstrate that TrustVLM’s ensemble score remains stable even under severely mismatched modality conditions.
> > > >
> > > >
> > > > **2. TrustVLM remains robust under imbalanced modality pairs.**
> > > >
> > > > We further assess scenarios where one modality is inherently more discriminative than the other. For each dataset, we measure the proportion of cases where the image-image similarity $S_{i-i}$ exceeds the image-text similarity $S_{i-t}$. This ratio directly reflects the relative discriminative strength of the visual vs. text modality. Table 3 below summarizes the results. The degree of imbalance varies substantially across datasets:
> > > >
> > > > For example, on Flowers102, we found that the $S_{i-i}$ provides a stronger discriminative signal than $S_{i-t}$ in 97.68% of cases. This directly explains why TrustVLM provides a substantial performance improvement on this dataset, achieving a 9.14% gain in AUROC over MSP.
> > > > On Cars, the $S_{i-t}$ is comparatively more informative. We found that the $S_{i-i}$ was more discriminative in only 42.99% of cases. Consequently, our method's improvement over the already-strong MSP baseline is smaller on this dataset.
> > > >
> > > > Despite these imbalances, TrustVLM consistently outperforms MSP and other baselines.
> > > >
> > > > |AUROC$\uparrow$|MSP|TrustVLM-D|ratio where $S_{i-i}$ > $S_{i-t}$|
> > > > |-|-|-|-|
> > > > Flower102|85.91|95.05|0.97|
> > > > DTD|79.81|88.55|0.83|
> > > > Aircraft|72.62|75.62|0.63|
> > > > Pets|89.94|90.05|0.70|
> > > > Caltech101|86.99|90.51|0.98|
> > > > Cars|81.95|82.05|0.42|
> > > > EuroSAT|76.39|85.48|0.83|
> > > > UCF101|85.98|90.21|0.91|
> > > > Food101|84.51|88.52|0.76|
> > > > SUN397|77.90|83.80|0.95|
> > > >
> > > > *Table 3. Relative discriminative strength of $S_{i-i}$ vs. $S_{i-t}$ across datasets, demonstrating imbalanced modality behavior.*

---

> > > > ### Author Response · Authors · 2025-11-28
> > > >
> > > > **3. Potential reasons behind the robustness.**
> > > >
> > > > **$S_{i-i}$ acts as a verification signal.** $S_{i-i}$ is explicitly conditioned on the VLM’s prediction $\hat{y}$:
> > > >
> > > > If $\hat{y}$ is wrong, then $E_{x}$​ is compared to the wrong class prototype $P_{\hat{y}}$, which is typically far in the embedding space and $S_{i-i}$ tends to be low.
> > > >
> > > > If $\hat{y}$ is correct, then $E_{x}$ is compared to the right prototype $P_{\hat{y}}$, which is close and $S_{i-i}$ tends to be high.
> > > >
> > > > Thus, $S_{i-i}$ behaves as a consistency check on $S_{i-t}$. When both $S_{i-i}$ and $S_{i-t}$ are high (Agreement), $\kappa$ increases and the prediction appears trustworthy. When one is high and the other is low (Disagreement), $\kappa$ decreases and the prediction is flagged as risky. This is quite different from ensembling arbitrary scores from unrelated modalities (like tactile–thermal), where there is no such *verify the predicted class* structure.
> > > >
> > > > **MisD metrics depend on ranking, not absolute values.** MisD metrics (AUROC, AURC, FPR@95) depend on relative ranking of $\kappa$ between correct and wrong predictions, not the absolute scale. As long as, on average, $\mathbb{E}[S_{i-i} \mid \text{correct}] \gtrsim \mathbb{E}[S_{i-i} \mid \text{wrong}]$, adding $S_{i-i}$​ to $S_{i-t}$ ​tends to improve or preserve the ranking. Correct predictions receive a positive contribution from $S_{i-i}$, while wrong predictions receive little or even a negative contribution. This widens the separation between the score distributions of correct and incorrect predictions, thereby improving MisD performance.
> > > >
> > > > **Averaging over prototypes mitigates severe mismatch.** Prototypes are averages over N-shot embeddings per class. Thus, even if some training images are noisy, or the auxiliary encoder has a domain bias, the class prototype $P_{c}$​ lies near the center of the class cluster, not at single outlier points. This means for a correct test image, similarity to $P_{\hat{y}}$​ is typically higher than to other prototypes, even under shift. For a wrong prediction, similarity to the wrong class prototype tends to be much lower. So even in the presence of moderate mismatch, $S_{i-i}$​ retains the *correct > wrong* tendency at the class level, which is exactly what the MisD ranking needs.
> > > > ___
> > > > We sincerely thank the reviewer again for the constructive feedback. If there are any additional questions or concerns, we would be more than happy to further address them.

---

> > > > > ### Comment · Reviewer_FUYk · 2025-11-28
> > > > > **Thank you for your detailed reply**
> > > > >
> > > > > Dear Authors,
> > > > >
> > > > > Thank you for your detailed feedback and comprehensive experiments! The results have addressed my major concern and it gives a clear understanding of the reason behind its robustness.
> > > > >
> > > > > I appreciate your efforts put into the rebuttal phase, and I think all reviewers are required to engage into the discussion just as they wanted others to do the same for their own papers.
> > > > >
> > > > > So far all my concerns are addressed and I would like to vote for acceptance of this paper.
> > > > >
> > > > > Best wishes,
> > > > > Reviewers.

---

> > > > > > ### Author Response · Authors · 2025-11-28
> > > > > > **Thank You for the Constructive Feedback and Acceptance Recommendation!**
> > > > > >
> > > > > > Dear Reviewer FUYk,
> > > > > >
> > > > > > Thank you very much for your positive feedback and for engaging in the discussion during the rebuttal phase! We completely agree with your point that all reviewers should actively participate in the discussion, just as they would expect others to do for their own papers. We value this collaborative process and believe it significantly improves the quality and clarity of the work.
> > > > > >
> > > > > > We are glad that our explanations and additional experiments have addressed your concerns. Following your suggestion, we have added a dedicated section in the revised manuscript (Appendix G) discussing the potential reasons behind the robustness of our proposed method.
> > > > > >
> > > > > > **Thank you once again for your constructive comments and for recommending our paper for acceptance!**
> > > > > >
> > > > > > Best regards,
> > > > > >
> > > > > > The Authors

---

### Official Review · Reviewer_CAb7 · 2025-11-01

**Soundness:** 3
**Presentation:** 3
**Contribution:** 2
**Rating:** 6
**Confidence:** 4

**Summary:**

In this work, the authors introduce TrustVLM, a training-free framework for predicting when a VLM’s predictions can be trusted. The task explored in this work is misclassification detection, which involves identifying when a prediction is incorrect. The key idea is to (1) generate visual prototypes for each class (i.e. the average embedding for N samples from the training data) and (2) compute the image-to-image similarity between the query image and the class prototypes. The standard image-to-text cosine similarity and the computed image-to-image similarity scores are combined in order to determine the overall prediction confidence. The authors show that this approach leads to substantially better misclassification detection performance than baselines.

**Strengths:**

- This work addresses an important task: determining when the predictions of a VLM are likely to be reliable.
- Although the proposed method is methodologically straightforward, strong performance is observed across a range of datasets and model backbones. The authors also compare with multiple baselines. The distribution shift experiments with ImageNet are particularly compelling.

**Weaknesses:**

- **Need for finer-grained analysis:** This paper could benefit from additional fine-grained analysis with respect to when the proposed method is most effective (rather than just overall metrics). For example, are there specific classes where misclassification detection performance improves substantially when using the proposed method (as compared to MSP)? What types of characteristics are common among those classes?
- **Variance of performance:** The proposed method is likely very sensitive to the choice of few-shot samples used to compose the class prototypes. What is the variance in performance when using prototypes composed from different randomly-selected N-shot sample sets?

**Questions:**

Questions are listed above under weaknesses.

---

> ### Author Response · Authors · 2025-11-19
>
> Thanks for your insightful reviews and great support of our paper! We provide the responses to your questions as follows:
>
> >**Q1**: Need for finer-grained analysis
>
> **A1**: Thank you for this insightful question. This fine-grained analysis is key to understanding our method, and we provide it in Appendix F (Table 15, Lines 910-965). The main finding is that TrustVLM is most effective when the classes are semantically similar but visually distinct. Our method's strength comes from fusing the VLM's semantic image-to-text similarity ($S_{i-t}$) and our visual image-to-image similarity ($S_{i-i}$). The relative benefit depends on which signal is more discriminative for a given dataset:
>
> **Visually Ambiguous, Semantically Distant Classes:** For objects that look similar but are semantically different (e.g., a "lemon" vs. a "tennis ball"), the VLM's semantic $S_{i-t}$ score is already highly discriminative (e.g., "fruit" vs. "sports object"). In these cases, the baseline MSP (which is based on $S_{i-t}$) performs well, and the gains from our $S_{i-i}$ score are modest.
>
> **Semantically Similar, Visually Distinct Classes:** This is precisely where TrustVLM excels. For fine-grained categories where text labels are very similar (e.g., species of flowers), the $S_{i-t}$ score often struggles. However, the subtle visual differences (e.g., petal shape) are effectively captured by our $S_{i-i}$ score, providing the missing discriminative power.
>
> We experimentally validated this hypothesis in Table 15 (and the table below) by directly comparing the discriminative power of the two modalities. For each image, we randomly sampled one positive example (same class) and one negative example (different class), and computed both image-to-image (i-i) and image-to-text (i-t) similarities. We then measured the ratio of cases where $S_{i-i}$ exceeds $S_{i-t}$. This ratio directly quantifies the relative discriminative strength of the visual features compared to the text-aligned semantic features.
>
> On Flowers102 (high semantic similarity), we found that the $S_{i-i}$ visual score provides a stronger discriminative signal than the $S_{i-t}$ semantic score in 97.68% of cases. This directly explains why TrustVLM provides a substantial performance improvement on this dataset, achieving a 9.14% gain in AUROC over MSP.
> On Cars, the $S_{i-t}$ semantic score is comparatively more informative. We found that the $S_{i-i}$ visual score was more discriminative in only 42.99% of cases. Consequently, our method's improvement over the already-strong MSP baseline is smaller on this dataset.
>
> In summary, the classes that benefit most from TrustVLM are those characterized by fine-grained visual distinctions that are not well-captured by the VLM's text-semantic alignment alone.
>
> ||MSP|TrustVLM-D|ratio where $S_{i-i}$ > $S_{i-t}$|
> |-|-|-|-|
> Flower102|85.91|95.05|0.97|
> DTD|79.81|88.55|0.83|
> Aircraft|72.62|75.62|0.63|
> Pets|89.94|90.05|0.70|
> Caltech101|86.99|90.51|0.98|
> Cars|81.95|82.05|0.42|
> EuroSAT|76.39|85.48|0.83|
> UCF101|85.98|90.21|0.91|
> Food101|84.51|88.52|0.76|
> SUN397|77.90|83.80|0.95|
> ___
> >**Q2**: Variance of performance
>
> **A2**: Thank you for this important question. We agree that evaluating the sensitivity to specific N-shot samples is an essential test of our method's robustness.
> We conducted this exact experiment, and the results are detailed in Tables 7 and 8. We ran tests on the Flowers102 dataset by constructing prototypes from three different, randomly-selected sets of images for both N=1 and N=4.
>
> The results show **extremely low variance**, demonstrating that TrustVLM is highly stable with respect to sample selection. For instance, with N=4 (Table 8), the AUROC across three random prototype sets is 94.80, 94.74, and 94.81, with a variance of 0.001. The FPR95 is 29.69, 29.07, and 29.81, with a variance of 0.158. The AURC is 78.15, 78.47, and 78.18, with a variance of 0.031.
>
> These negligible variances confirm that TrustVLM’s performance is highly consistent regardless of which specific few-shot samples are used to construct the prototypes. This robustness further demonstrates that the method does not rely on carefully curated reference examples and remains reliable even under random sampling.
> ___
> We sincerely thank the reviewer again for the constructive feedback. If there are any additional questions or concerns, we would be more than happy to further address them.

---

### Official Review · Reviewer_pKks · 2025-11-01

**Soundness:** 3
**Presentation:** 4
**Contribution:** 2
**Rating:** 6
**Confidence:** 4

**Summary:**

For improving the reliability of VLM, this paper proposes a new training-free framework TrustVLM to estimate when VLM’s prediction will be trusted. The key of TrustVLM is to use a new confidence scoring function which will use not only cosine similarity between image and text but also the addition information from image embedding space. This additional information is the image-to-image relations or similarity. The idea is very straightforward where you need to extract some prior knowledge from training data. This prior knowledge is the class embeddings that are extracted from N shot examples from training data for each class. It likes you did another way for image classification based on the similarity between class embeddings and input image embedding. The whole idea is clear. The paper is well written. The key question here is you need to have a training data and get the class embedding based on the classes in the training data. But in some use cases, we don’t have the training data to get these class embeddings in advance. The proposed method has its limitation on it.

**Strengths:**

1.	TrustVLM is a training-free framework designed to evaluate the reliability of VLM predictions. One of its key advantages is that it does not require additional training, which makes it convenient to apply in scenarios where labeled data is limited or unavailable. The framework combines both image-to-text and image-to-image similarities, which allows for a more robust and nuanced design of confidence scores. This combination provides a richer representation of the visual information, enabling the framework to better capture the model’s uncertainty.

2.	The paper demonstrates that the proposed visual prototypes not only enable more reliable confidence estimation but also enhance fine-grained classification accuracy.

3.	The experiments conducted across diverse datasets, model architectures, and VLMs show the generality and effectiveness of TrustVLM.

**Weaknesses:**

1.	A notable limitation of this method is that it relies on the availability of in-domain data that includes images for all classes to be predicted. Under this assumption, the method can extract and store visual prototypes for each class, which are then used for confidence estimation. However, in many practical scenarios, obtaining such in-domain data for every class may be difficult or infeasible. Moreover, if the training or reference data does not fully cover the diversity of the test data, the method may encounter out-of-distribution (OOD) situations. As the introduction clearly states, TrustVLM is not designed to handle OOD cases, which inherently limits its applicability in environments where data coverage is incomplete or classes are highly dynamic. This restriction should be carefully considered when evaluating the practical utility of the method.

2.	There exist alternative strategies to improve reliability when prior knowledge about class representations is available. For instance, one could employ a separate image embedding model to independently obtain embeddings for the input and for each class, then compute similarity scores between them. Another potential approach is to first predict the class of the image using a preliminary classifier and then use this prediction as an input for the final classification task. These alternatives might offer comparable or complementary benefits to TrustVLM. Therefore, the paper would be strengthened if the authors could provide a more detailed comparison or discussion of how TrustVLM differs from these approaches. Specifically, it would be helpful to clarify the unique advantages of TrustVLM, such as why its combined use of image-to-text and image-to-image similarity provides superior or more reliable confidence estimation compared to these other methods. This explanation would help to more clearly establish the method’s contributions and practical significance.

**Questions:**

Please check the weaknesses.

---

> ### Author Response · Authors · 2025-11-19
>
> Thanks for your insightful reviews and great support of our paper! We provide the responses to your questions as follows:
>
> >**Q1**: Dependence on Complete In-Domain Data and Lack of OOD Handling
>
> **A1**: We thank the reviewer for the thoughtful feedback. We agree that the reliance on reference data is an important aspect to clarify. TrustVLM is designed for in-distribution (ID) misclassification detection (MisD), which is fundamentally different from out-of-distribution (OOD) detection (Lines 052–078). For MisD, assuming access to a small reference set of known classes is standard and often necessary. Nonetheless, we emphasize that this requirement is far less restrictive in practice than it may initially appear. Our paper demonstrates this across three dimensions:
>
> **1. Extreme Data Efficiency:** The reviewer's concern about the difficulty of obtaining data is highly valid. This is precisely why we evaluated TrustVLM in extreme few-shot settings. As shown in Fig. 8, TrustVLM outperforms a strong MSP baseline even with only one labeled example per class (N=1). This highlights that our method does not require a large or diverse reference dataset, just a single representative prototype per class. Moreover, Tables 7 and 8 show that performance is stable across different prototype choices.
>
> **2. Demonstrated Robustness under Distribution Shifts:** The reviewer is correct that limited reference diversity could lead to OOD situations. We explicitly tested this scenario. In Fig. 3 and Table 10, we construct prototypes using only ImageNet training images and evaluate directly on challenging shifted variants (ImageNet-A, -R, -Sketch, -V2). TrustVLM consistently outperforms baselines under these shifts, demonstrating strong robustness even when the reference data does not fully reflect test-time variability. When diversity is a concern, TrustVLM also enables an efficient alternative: treating prototypes as learnable parameters and fine-tuning them on a small N-shot set, which further improves performance with negligible cost.
>
> **3. Feasibility Even with No Real Data:** In the most extreme case where collecting any in-domain data is infeasible (e.g., due to privacy), we demonstrate a viable alternative. In Appendix E (Table 9, Lines 852-863), we show that using synthetically generated images from a generative model (Stable Diffusion 3) to create the prototypes still outperforms the MSP baseline. This provides a practical path for deployment even in data-scarce or private settings.
>
> In summary, while TrustVLM leverages reference prototypes, this requirement is highly practical. Our method is **effective with just one example per class, remains robust under significant distribution shifts, and can even operate using purely synthetic data**. We believe these results collectively demonstrate that this data dependency is not a limiting factor for its practical utility.

---

> ### Author Response · Authors · 2025-11-19
>
> >**Q2**: how TrustVLM's unique combination of similarity metrics offers superior reliability and distinct advantages
>
> **A2**: We thank the reviewer for raising this insightful question, as it perfectly frames the core design choices and contributions of TrustVLM. The two alternatives suggested: (1) using only image-to-image (i-i) similarity and (2) using a preliminary prediction to guide a final decision, are indeed natural baselines. Our experiments were designed precisely to evaluate these alternatives and demonstrate our method's unique advantages.
>
> **1. Superiority of Combined (i-t + i-i) vs. (i-i only):** The reviewer's first alternative, relying solely on a separate vision encoder for i-i similarity, is exactly what we evaluate as a unimodal baseline in our ablation study in Table 6.
> Relying only on image-to-text similarity (i-t, the MSP baseline) is suboptimal. Relying only on image-to-image similarity (i-i) is also suboptimal. Our full method, which combines both, consistently achieves the best performance. Thus, the ablation clearly demonstrates that neither component is sufficient alone; their combination is necessary for optimal reliability.
>
> **2. The Unique Advantage of Complementarity:** The key reason TrustVLM outperforms the alternatives is that i–t and i–i capture fundamentally different and complementary cues. Appendix F (Lines 910-965) provides an in-depth analysis, and we summarize the intuition here:
>
> **When i-t excels:** Consider visually similar but semantically distinct classes, like "lemon" and "tennis ball". A purely visual $S_{i-i}$ comparison might find them highly similar. However, the VLM's semantic $S_{i-t}$ score can easily distinguish "a sour fruit" from "a sports object".
>
> **When i-i excels:** Consider fine-grained classes with subtle visual differences, like different species of flowers (e.g., Flowers102 dataset). The VLM's $S_{i-t}$ score may be very close for two species. In this case, the $S_{i-i}$ score, derived from a strong vision encoder, provides the critical, subtle visual discrimination that the text-aligned features miss.
>
> Our quantitative analysis in Table 15 validates this: on Flowers102, the $S_{i-i}$ score is more discriminative in 97.68\% of cases, whereas on Cars, this drops to 42.99\% (meaning $S_{i-t}$ is more informative).  TrustVLM’s unique advantage is that it fuses both, leveraging the VLM's rich semantic knowledge (i-t) and the vision encoder's specialized visual discrimination (i-i) for a more robust confidence score.
>
> **3. Relation to a Preliminary Classifier:** The reviewer's second suggestion on using a preliminary classifier to guide the task is precisely how TrustVLM is designed, as shown in Figure 2. We use the VLM's standard zero-shot output (Eq. 1) as the preliminary prediction $\hat{y}$. This prediction is then used to select the corresponding visual prototype $P_{\hat{y}}$ for verification. Our key innovation is that we don't treat this as a separate task. Instead, we compute a verification score ($S_{i-i}$) and fuse it with the original confidence score ($S_{i-t}$) to produce our final, more reliable metric $\kappa(x)$.
>
> In summary, TrustVLM's design integrates the very components the reviewer suggested. Our ablations (Table 6) and fine-grained analysis (Appendix F) empirically demonstrate that it is the fusion of these complementary i-t and i-i signals, rather than the use of either in isolation, that constitutes the method's core contribution and provides superior reliability.
> ___
> We sincerely thank the reviewer again for the constructive feedback. If there are any additional questions or concerns, we would be more than happy to further address them.

---

### Meta-Review · Area_Chair_rMFZ · 2026-01-05

**Summary:**

This paper proposes a misclassification detection framework for vision-language models (VLMs), such as CLIP or SigLIP. The proposed method, named TrustVLM, uses additional image-to-image similarities; while the zero-shot approaches use text-to-image similarity, where text is the class name, this method proposes to additionally use images for each class as a KNN classifier. The final confidence score is computed by combining both similarities. Experimental results show that the proposed method performs better than baseline methods in AURC, AUROC, and FPR@95.

**Reviewer Concerns:**

This paper has mixed initial opinions with three borderlines and one rejection. There were several concerns raised by the reviewers, and some of them were (partially) resolved by the rebuttal comments. For example,

- The method requires pre-collected image samples for each label, which may limit practical deployment (pKks, aUQM, FUYk).
    - The rebuttal comment showed that the proposed method works well even with the N=1 case. Additionally, the rebuttal comment clarified that the proposed method still works with synthetic images by StableDiffusion.
    - However, this argument still looks valid to the AC. The detailed comment can be found below.
- Lack of robustness analyses under misalignment, distribution shift (FUYk, CAb7)
    - The rebuttal clarified that the role of i-i is a verification and supported this with results on ImageNet variants. One of the reviewers explicitly stated that the concern was resolved.
- Lack of deeper analyses (CAb7)
    - The rebuttal comment added analysis suggesting large gains for semantically similar but visually distinct cases.
    - The rebuttal comment clarified i-t vs i-i results.

However, I think that a few concerns still potentially remain after the rebuttal comment.

- The proposed method still needs image samples (pKks, aUQM, FUYk). Although the rebuttal experiments showed that the proposed method works with generated images with StableDiffusion. However, StableDiffusion is trained on web images, where ImageNet is also collected from the web. In other words, if the images are very different from the web images or very different from the generated images, the method is not guaranteed to work well. This argument can be supported by Table 10: when the distribution gap becomes larger (ImageNet-A, ImageNet-R), the gap between the baseline methods and the proposed method becomes very small.
- Novelty issue (aUQM, pKks). As pointed out by the reviewers, the proposed method looks like a straightforward combination of known ingredients. Even though the paper described its motivation as a modality gap between image and text, the AC thinks that this is a combination (or ensemble) of an image KNN classifier and a text-based zero-shot classifier. To me, the performance gain seems to come from the ensemble, not necessarily from the proposed i-t and i-i combinations.

Overall, I think that this paper solves the misclassification detection problem of zero-shot classification with CLIP-like models by a simple ensemble method. Despite its empirical gains in some benchmarks, there are a few results that show the fundamental limitation of the proposed method (e.g., the proposed method does not outperform the baselines on ImageNet-A and ImageNet-R, where the distribution gap is larger). I think that the advantage of this paper (empirical gains in many datasets and simplicity) does not exceed its disadvantages (the method fundamentally needs "images" regardless of real or generated, the method fundamentally cannot deal with out-of-distribution images, and the novelty of the method is somewhat limited). Therefore, I recommend rejection for this paper.

**Reviewer Scores:**

Three reviewers initially voted for borderline. One reviewer explicitly mentioned that their major concerns were resolved. I also think that the other two reviewers could revise their opinions if they actively engaged in the discussion between the authors.

However, I don't think the reviewer who initially voted for rejection would change their opinion after the discussion period. The reviewer participated in the discussion and clarified why they thought that the novelty is limited by suggesting four related works (two few-shot clip adaptation and two clip + external vision encoders). I generally agree with the reviewer's opinion on the novelty issue.

---

### Decision · Program_Chairs · 2026-01-26

Reject